# Security analysis and secure channel-free certificateless searchable public key authenticated encryption for a cloud-based Internet of things

**Bin Wu**[1,2], **Caifen Wang**[3]*, **Hailong Yao**[1,4]

**1** College of Mathematics and Statistics, Northwest Normal University, Lanzhou, China, **2** Information Security Lab, Lanzhou Resources and Environment Voc-tech College, Lanzhou, China, **3** College of Big Data and Internet, Shenzhen Technology University, Shenzhen, China, **4** School of Electronic and Information Engineering, Lanzhou City University, Lanzhou, China

* soloren@yeah.net

**Data Availability Statement:** All relevant data are within the paper.

**Funding:** The author(s) received no specific funding for this work.

## Abstract

With the rapid development of informatization, an increasing number of industries and organizations outsource their data to cloud servers, to avoid the cost of local data management and to share data. For example, industrial Internet of things systems and mobile healthcare systems rely on cloud computing's powerful data storage and processing capabilities to address the storage, provision, and maintenance of massive amounts of industrial and medical data. One of the major challenges facing cloud-based storage environments is how to ensure the confidentiality and security of outsourced sensitive data. To mitigate these issues, He et al. and Ma et al. have recently independently proposed two certificateless public key searchable encryption schemes. In this paper, we analyze the security of these two schemes and show that the reduction proof of He et al.'s CLPAEKS scheme is incorrect, and that Ma et al.'s CLPEKS scheme is not secure against keyword guessing attacks. We then propose a channel-free certificateless searchable public key authenticated encryption (dCLPAEKS) scheme and prove that it is secure against inside keyword guessing attacks under the enhanced security model. Compared with other certificateless public key searchable encryption schemes, this scheme has higher security and comparable efficiency.

## Introduction

The Internet of things (IoT) [1–3] is a new model that has rapidly become popular in wireless communication scenarios. The basic idea of this concept is that all items—such as actuators, radio-frequency identification tags—are connected to the Internet through information sensing devices, to exchange information. That is, objects are interconnected, to realize intelligent identification and management. IoT has opened new avenues for technology connectivity and business upgrading in industry, healthcare, and transportation, of which the industrial Internet of things (IIoT) and mobile healthcare systems (MHSs) are the most successful applications.

**Competing interests:** The authors have declared that no competing interests exist.

IIoT refers to the IoT environment applied in industrial systems. IIoT integrates various intelligent terminals and sensing devices through a ubiquitous network to efficiently and economically manage industrial production, not only improving manufacturing efficiency, but also reducing product costs, upgrading traditional industries to intelligent industries [4].

MHSs refers to the provision of medical applications and medical information at any time or place, based on the IoT [5, 6]. MHSs provide a wide range of services and applications, including patient monitoring, mobile telemedicine, real-time transmission, storage of (and access to) medical information, and customized and personalized medical service prescriptions.

Although the IIoT and MHSs have great development prospects and bring great convenience to people's productivity and life, they still face a substantial challenge, namely, the storage and management of massive amounts of data (including both industrial and medical data).

In recent years, cloud computing technology has developed rapidly, and some typical cloud service products have been released and have received extensive attention; these include Dropbox [7], a cloud network storage tool, and Windows Azure [8], a cloud computing platform from Microsoft. Cloud computing is a business model that allows on-demand network access to configurable computing resources such as services, storage, networks, and applications. These resources can be quickly provided and released with minimal management work and interaction. The IoT generally contains small objects (things) with limited processing power and storage capabilities, whereas cloud computing has unlimited storage and processing power capabilities, which can play a supporting role in the IoT architecture, as explained in Ref. [9, 10]. Specifically, the IoT can benefit from the unlimited resources and capabilities of the cloud to make up for its technological constraints. A recent and continuing trend is the integration of the cloud and the IoT. The new model, called the cloud-based Internet of Things, has been extensively studied [11–14]. In a cloud-based IoT system, users upload data collected by various smart devices to cloud servers through the Internet, and other authorized users can retrieve data collected from different environments.

However, when outsourcing data to a cloud server, the security and privacy of the data cannot be guaranteed because the cloud server is considered honest but curious; it can fulfill its obligations, but is curious about the stored information. Before uploading sensitive data to the cloud server, the data owner needs to perform encryption to protect the privacy and confidentiality of the data. However, in this way, the existing plaintext-based keyword search technology is ineffective, because encryption usually hides the structure of the original data. To address this problem, searchable encryption (SE), which supports efficient search over ciphertext, has been widely applied, studied, and developed in recent years [15–29].

SE can be categorized into symmetric and asymmetric encryption. Symmetric searchable encryption has the characteristics of low computational overhead and high speed, but it is usually suitable for a single-user model; additionally, the encryption and decryption parties need to negotiate the key beforehand. To address this limitation, public key searchable encryption (PEKS) was first proposed by Boneh et al. [18]. It is very suitable for solving the searchable encryption problem in a multi-user system. In a PEKS system, without prior agreement between sender and receiver, the sender generates encrypted files, called PEKS ciphertext (including encrypted files body and encrypted keywords) using the receiver's public key, and uploads the ciphertext to the cloud server. When the receiver needs to search the ciphertext for a certain keyword, it uses its own secret key to generate the search certificate of the keyword and sends it to the cloud server. The server then runs a test operation to select the ciphertext file containing the target keyword, and returns it to the receiver.

Although PEKS solves the problem of searching ciphertext, it still suffers from some privacy problems. Reference [24] pointed out that most PEKS schemes are susceptible to off-line

keyword guessing attacks (KGAs). The KGAs is attributed to the fact that keyword space is very small and users usually use common keywords for retrieval, which provides a "shortcut" for an attacker to obtain data privacy information by using only dictionary attacks. Specifically, with a given trapdoor, the attacker tests every possible keyword off-line. If the test is successful, the attacker can know the potential keywords in the trapdoor. From the server's reply, he also knows which encrypted files contain the keywords encapsulated in the trapdoor. In short, by running this off-line KGAs, malicious (inside or outside) attackers can obtain information about encrypted files and invade the user's data privacy. Constructing a scheme to resist KGAs has attracted the attention of many researchers [30–37].

Recently, He et al. [38] proposed a new scheme, CLPAEKS, for IIoT, and Ma et al. [39] proposed a scheme, CLPEKS, for MHSs. Their schemes are both certificateless public key searchable encryption schemes, which effectively solve the problem of searching over encrypted data stored in the cloud and avoid the problems of certificate management and key escrow.

In this paper, through careful analysis, we describe security vulnerabilities that we found in the two schemes mentioned above. The security reduction of He et al.'s CLPAEKS scheme is actually incorrect for two types of adversaries. That is, an adversary cannot solve the computational bilinear Diffie-Hellman problem by using adversary $\mathcal{A}_{\mathcal{I}}$ ($\mathcal{A}_{\mathcal{II}}$), which attacks the security of the CLPAEKS scheme, as a subroutine. Ma et al.'s CLPEKS scheme is not secure against off-line KGAs. Furthermore, in both CLPAEKS and CLPEKS, anyone can run test operations, which makes it easy to identify whether two search queries are generated from the same keyword; that is, the search patterns of users can be revealed to anyone. The potential risks of search pattern leakage have been studied in the literature [40]: adversaries may use searching frequency to obtain information about the plaintext.

## Our contributions

- We note the security vulnerabilities of the CLPAEKS scheme proposed by He et al. and the CLPEKS scheme proposed by Ma et al.

- To protect the privacy and security of data stored in the cloud in the Internet of Things environment, we propose a dCLPAEKS scheme, which is a channel-free certificateless searchable public key encryption scheme, and present a security model for dCLPAEKS to remedy the problem mentioned above.

- Under the enhanced security model, we prove that the dCLPAEKS scheme is secure against inside keyword guessing attacks for two types of adversaries. Specifically, we formally prove that the scheme satisfies ciphertext indistinguishability and trapdoor indistinguishability. Furthermore, we prove that the scheme satisfies the security of the designated tester (specifically, only the server can perform test operations).

- We compare our scheme with other CLPEKS schemes in terms of security, computational complexity and communication overhead. We also evaluate its efficiency in experiments, and the results show that our scheme has higher security and efficiency.

## Related works

In 2004, Boneh et al. first proposed the concept of public key searchable encryption [18] and proposed the construction scheme of PEKS based on anonymous identity-based cryptosystems. This scheme has been applied in a mail system to solve the mail routing problem of untrustworthy servers. In 2008, Baek et al. [23] pointed out that the scheme in [18] must be built on a secure channel. To overcome this limitation, they proposed a public key

searchable encryption scheme without a secure channel by introducing a designated tester. In 2006, Bynn et al. [30] found that the scheme proposed in [18] was susceptible to off-line KGAs because keywords are selected from a much smaller space than keys and users usually use common keywords; hence, an attacker can easily crack the PEKS system through KGAs. To protect against KGAs, Rhee et al. [32] proposed a trapdoor secure dPEKS scheme, but Wang et al. [41] later pointed out that the scheme suffered from an inherent insecurity, namely, vulnerability to inside KGAs (IKGAs). Roughly speaking, given a trapdoor, a malicious server can generate the PEKS ciphertext for any keyword it chooses, and then the server can run a test operation to determine whether the keywords being guessed are the keywords underlying the trapdoor. In 2013, Xu et al. [34] proposed a fuzzy keyword public key searchable encryption scheme against IKGAs. In their scheme, the server can only perform fuzzy matching search, and accurate matching search is executed locally, so an attacker cannot obtain an accurate search trapdoor, thus ensuring the security of the scheme. In 2016, Chen et al. [35] proposed a dual-server public-key searchable encryption scheme that can resist IKGAs from malicious servers by dividing the test algorithm into two parts and letting two independent servers execute it. However, all of the schemes mentioned above encounter certificate management or key escrow problems. To address this problem, AL-piyami et al. [42] defined the concept of certificateless public key cryptography (CLPKC). Users' private keys in certificateless public key cryptosystems consist of two parts: one is generated by the key generation center and the other is generated by users. Peng [43] proposed the first certificateless public key encryption with keyword search (CLPEKS) scheme. Subsequently, other improved certificateless public key searchable encryption schemes were proposed [44, 39]. He et al. [38] demonstrated that these schemes are vulnerable to IKGAs and proposed a certificateless public key authenticated encryption scheme with keyword search that can resist IKGAs.

## Paper organization

The rest of this paper is organized as follows. In section 2, we present some preliminaries. In section 3, we review Ha et al's scheme and Ma et al's scheme and then point out the disadvantages of their schemes. In section 4, we introduce a new notion, dCLPAEKS, and give its security model. We construct a concrete dCLPAEKS scheme and prove its security in enhanced security models in section 5. In section 6, we present the performance analysis of our proposed scheme. Finally, we conclude the paper in section 7.

## Preliminaries

### Bilinear pairing

Bilinear pairing [45] plays an important role in constructing many cryptographic schemes, including our dCLPAKES scheme. Let $\mathbb{Z}$ be a set of integers. Set $\hat{e} : \mathbb{G}_1 \times \mathbb{G}_1 \to \mathbb{G}_2$ as a bilinear map, mapping groups $\mathbb{G}_1$ and $\mathbb{G}_1$ to $\mathbb{G}_2$, where $\mathbb{G}_1$, $\mathbb{G}_2$ are cyclic groups with the same prime order $p$. This mapping satisfies the following properties:

1. **Bilinearity**: For any $u, v \in \mathbb{Z}_p^*$ and $g, h \in \mathbb{G}_1$, $\hat{e}(g^u, h^v) = \hat{e}(g, h)^{uv}$.

2. **Non-degeneracy**: If $g$ is a generator of $\mathbb{G}_1$, then $\hat{e}(g, g)$ is a generator of $\mathbb{G}_2$.

3. **Computability**: For any $g, h \in \mathbb{G}_1$, there is an efficient algorithm to calculate $\hat{e}(g, h)$.

## Computational bilinear diffie-hellman problem

**Definition 1** *(CBDH Problem) Let $\hat{e} : \mathbb{G}_1 \times \mathbb{G}_1 \to \mathbb{G}_2$ be a bilinear pairing. Given $(g, g^a, g^b, g^c)$, where $a, b, c \in \mathbb{Z}_p^*$ are unknown numbers, the goal is to compute the value of $\hat{e}(g, g)^{abc}$.*

## Decisional bilinear diffie-hellman assumption

The decisional bilinear Diffie-Hellman (DBDH) problem is described as follows.

Given $\mathsf{Y} = (g, g^x, g^y, g^z \in \mathbb{G}_1, \hat{e} : \mathbb{G}_1 \times \mathbb{G}_1 \to \mathbb{G}_2, Z \in \mathbb{G}_2)$, where $x, y, z$ are randomly chosen from $\mathbb{Z}_p^*$, let $\eta$ be a bit such that $\eta = 1$ if $Z$ is randomly selected from $\mathbb{G}_2$, and $\eta = 0$ if $Z = \hat{e}(g, g)^{xyz}$. The DBDH problem is to determine the value of $\eta$.

**Definition 2** *(DBDH Assumption [46, 47]) The DBDH assumption is that for any probabilistic polynomial-time (PPT) algorithm $\mathcal{A}$, the following holds*:

$$|\Pr[0 \leftarrow \mathcal{A}(\mathsf{Y})|\eta = 0] - \Pr[0 \leftarrow \mathcal{A}(\mathsf{Y})|\eta = 1]| \leq negl(\lambda)$$

*where the probability is taken over the random choice of $x, y, z \in \mathbb{Z}_p^*, g \in \mathbb{G}_1, Z \in \mathbb{G}_2$.*

# Review and security analysis of the CLPAEKS and CLPEKS schemes

In this section, we briefly review the CLPAEKS scheme of He et al. [38] and the CLPEKS scheme of Ma et al. [39], and give the security cryptanalysis of the two schemes.

## Review and security analysis of He et al.'s CLPAEKS scheme

**Description of He et al.'s scheme.** The CLPAEKS scheme can be described as follows:

- **Setup**: Input a security parameter $l$. The KGC selects two cyclic groups $\mathbb{G}_1, \mathbb{G}_2$ with the same prime order $q$ and a bilinear pairing $\hat{e} : \mathbb{G}_1 \times \mathbb{G}_1 \to \mathbb{G}_2$. Let $P$ be a generator of $\mathbb{G}_1$; The KGC chooses a random number $s \in Z_q^*$ as the master key and computes $P_{pub} = sP$. The KGC selects three different hash functions: $h_1 : \{0, 1\}^* \times \mathbb{G}_1 \to Z_q^*$, $H_2 : \{0, 1\}^* \to \mathbb{G}_1$ and $h_3 : \{0, 1\}^* \times \mathbb{G}_1 \times \mathbb{G}_1 \times \mathbb{G}_1 \to Z_q^*$. Then, the KGC publishes the system parameters $prms = \{l, \mathbb{G}_1, \mathbb{G}_2, \hat{e}, P, P_{pub}, h_1, H_2, h_3\}$.

- **Extract-Partial-Private-Key**: Input the sender's identity $ID_S \in \{0, 1\}^*$. The KGC selects a random number $r_{ID_S} \in Z_q^*$ and computes $R_{ID_S} = r_{ID_S}P$, $\alpha_{ID_S} = h_1(ID_S, R_{ID_S})$ and $d_{ID_S} = r_{ID_S} + s\alpha_{ID_S}(mod \ q)$. Then, the KGC returns $(d_{ID_S}$ and $R_{ID_S})$ to the sender. In parallel, the partial private key $(d_{ID_R}, R_{ID_R})$ of the receiver is calculated in the same way.

- **Set-Secret-Value**: This takes $ID_S, ID_R \in \{0, 1\}^*$ as input. The sender and the receiver choose random numbers $x_{ID_S}$ and $x_{ID_R}$ as their secret values, respectively.

- **Set-Private-Key**: This sets the sender's private key and the receiver's private key as $SK_{ID_S} = (x_{ID_S}, d_{ID_S})$ and $SK_{ID_R} = (x_{ID_R}, d_{ID_R})$, respectively.

- **Set-Public-Key**: The sender computes $P_{ID_S} = x_{ID_S}P$ and sets $PK_{ID_S} = (P_{ID_S}, R_{ID_S})$ as its public key. The receiver computes $P_{ID_R} = x_{ID_R}P$ and sets $PK_{ID_R} = (P_{ID_R}, R_{ID_R})$ as its public key.

- **CLPAEKS**: This takes $prms, ID_S, ID_R, SK_{ID_S}, PK_{ID_R}$ as input. The sender encrypts the keyword $w$ as follows:

  1. The sender chooses a random number $r \in Z_q^*$.

2. The sender computes $\beta_{ID_S} = h_3(ID_S, P_{pub}, P_{ID_S}, R_{ID_S})$, $\beta_{ID_R} = h_3(ID_R, P_{pub}, P_{ID_R}, R_{ID_R})$, $C_1 = (d_{ID_S} + \beta_{ID_S} x_{ID_S})H_2(w) + rP$, $C_2 = r(\beta_{ID_R} P_{ID_R} + R_{ID_R} + \alpha_{ID_R} P_{pub})$.
The final ciphertext for the keyword is $C = (C_1, C_2)$.

- **Trapdoor**: This takes $prms, ID_S, ID_R, SK_{ID_R}, PK_{ID_S}$ as input. The data receiver runs the following steps to compute the trapdoor $T_w$:

  1. Compute $\beta_{ID_R} = h_3(ID_R, P_{pub}, P_{ID_R}, R_{ID_R})$, $\beta_{ID_S} = h_3(ID_S, P_{pub}, P_{ID_S}, R_{ID_S})$.

  2. Compute $T_w = \hat{e}((d_{ID_R} + \beta_{ID_R} x_{ID_R})H_2(w), \beta_{ID_S} P_{ID_S} + R_{ID_S} + \alpha_{ID_S} P_{pub})$.

- **Test**: Take $prms$, the trapdoor $T_w$ and ciphertext $C$ as input. The cloud server checks whether $T_w \hat{e}(C_2, P) = \hat{e}(C_1, \beta_{ID_R} P_{ID_R} + R_{ID_R} + \alpha_{ID_R} P_{pub})$ holds. If it holds, then the server outputs 1. Otherwise, it outputs 0.

**Security analysis.** In the random oracle model, the semantic security of the CLPAEKS scheme against IKGAs is reduced to solve the CBDH problem [38]. Here, we show that the security reduction for the CLPAEKS scheme is in fact incorrect for two types of adversaries. We use a reductionist proof for a type 1 adversary as an example to illustrate.

Given an instance $(P, aP, bP, cP)$ of the CBDH problem, assuming that adversary $\mathcal{A}_I$ intends to break the CLPAEKS scheme, $\mathcal{B}$ makes use of the advantage of $\mathcal{A}_I$ to compute the value of $\hat{e}(P, P)^{abc}$. $\mathcal{B}$ simulates security games for adversary $\mathcal{A}_I$. After $\mathcal{A}_I$ outputs a guess value in the guess stage, the simulator $\mathcal{B}$ calculates $\hat{e}(P, P)^{abc}$ as follows:

$$\frac{\hat{e}(d_{ID_I}P + \beta_I cP, \beta_{ID_S} P_{ID_S} + R_{ID_S} + \alpha_{ID_S} aP)^{b+\mu_i}}{\hat{e}(d_{ID_I}P + \beta_I cP, \beta_{ID_S} P_{ID_S} + R_{ID_S} + \alpha_{ID_S} aP)^{\mu_i}}$$
$$= \hat{e}(d_{ID_I}P + \beta_I cP, \beta_{ID_S} P_{ID_S} + R_{ID_S} + \alpha_{ID_S} aP)^{b}$$

$$\frac{\hat{e}(d_{ID_I}P + \beta_I cP, \beta_{ID_S} P_{ID_S} + R_{ID_S} + \alpha_{ID_S} aP)^{b}}{\hat{e}(d_{ID_I}bP, \beta_{ID_S} P_{ID_S} + R_{ID_S} + \alpha_{ID_S} aP)}$$
$$= \hat{e}(\beta_I cP, \beta_{ID_S} P_{ID_S} + R_{ID_S} + \alpha_{ID_S} aP)^{b}$$

$$\left(\frac{\hat{e}(\beta_I cP, \beta_{ID_S} P_{ID_S} + R_{ID_S} + \alpha_{ID_S} aP)^{b}}{\hat{e}(\beta_I cP, \beta_{ID_S} x_{ID_S} bP + r_{ID_S} bP)}\right)^{\frac{1}{\beta_I \alpha_{ID_S}}}$$
$$= \hat{e}(P, P)^{abc}$$

The core part of computing $\hat{e}(P, P)^{abc}$ is the left-hand side of the first equation. For ease of description, the numerator and denominator of the fraction in the first equation are abbreviated as follows:

$$\mathbb{M} = \hat{e}(d_{ID_I}P + \beta_I cP, \beta_{ID_S} P_{ID_S} + R_{ID_S} + \alpha_{ID_S} aP)^{b+\mu_i}$$
$$\mathbb{N} = \hat{e}(d_{ID_I}P + \beta_I cP, \beta_{ID_S} P_{ID_S} + R_{ID_S} + \alpha_{ID_S} aP)^{\mu_i}$$

Let us see how $\mathcal{B}$ obtains $\mathbb{M}$ and $\mathbb{N}$.

$\mathbb{N}$ is calculated by $\mathcal{B}$ itself, while $\mathbb{M}$ is obtained by $\mathcal{B}$ using $\mathcal{A}_I$. However, $\mathcal{B}$ is unable to use the adversary $\mathcal{A}_I$ to obtain the value of $\mathbb{M}$. Specifically, in the reductionist proof for adversary $\mathcal{A}_I$, $\mathbb{M}$ is the value of the trapdoor of the challenge keywords under the challenge identities

and $E_5$ denotes the event that $\mathcal{A}_\mathcal{I}$ does not ask the hash query for the value of the trapdoor of the challenge keywords, and we show that $Pr[\neg E_5] \geq 2\varepsilon$, where $\varepsilon$ denotes the advantage of $\mathcal{A}_\mathcal{I}$ breaking the CLPAEKS scheme. Thus, $\mathcal{B}$ aims to make use of the fact that $\mathcal{A}_\mathcal{I}$ has conducted a hash query on this trapdoor, namely, the value of $\mathbb{M}$, with a non-negligible probability in the interactive game, and then randomly select one from the previous query history as $\mathbb{M}$.

However, in the trapdoor algorithm design of the CLPAEKS scheme [38], no hashing operation is performed on the value of the trapdoor, that is, $H_i(T_w)$, for some hash function $H_i$. Therefore, it is impossible for $\mathcal{A}_\mathcal{I}$ to make the hash query on the trapdoor of the challenge keyword. In addition, $\mathcal{B}$ has no other advantage in obtaining $\mathbb{M}$ from $\mathcal{A}_\mathcal{I}$. In short, the reduction process shows that $\mathcal{B}$ is unable to solve the CBDH problem with $\mathcal{A}_\mathcal{I}$ as a subroutine.

## Review and security analysis of Ma et al.'s CLPEKS scheme

**A description of Ma et al.'s scheme.**  The CLPEKS scheme is as follows:

- **Setup**: Input a security parameter $k$. The KGC selects two cyclic groups $\mathbb{G}_1$, $\mathbb{G}_2$ with the same prime order $q$, a bilinear pairing $e : \mathbb{G}_1 \times \mathbb{G}_1 \to \mathbb{G}_2$. Let $P$ be a generator of $\mathbb{G}_1$; the KGC chooses a random number $s \in Z_q^*$ as the master key and computes $P_{pub} = sP \in \mathbb{G}_1$. The KGC selects four different hash functions: $h_1 : \{0,1\}^* \times \mathbb{G}_1 \to Z_q^*$,
  $h_2 : \{0,1\}^* \times \mathbb{G}_1 \times \mathbb{G}_1 \to Z_q^*$, $H_3 : \{0,1\}^* \to \mathbb{G}_1$ and $h_4 : \mathbb{G}_1 \to \{0,1\}^l$. Then, the KGC publishes public parameters $prms = \{k, \mathbb{G}_1, \mathbb{G}_2, \hat{e}, q, P, P_{pub}, h_1, h_2, H_3, h_4\}$.

- **Extract-Partial-Private-Key**: Input a user U's identity $ID \in \{0,1\}^*$. The KGC selects a random number $t_{ID} \in Z_q^*$ and computes $T_{ID} = t_{ID}P$, $\alpha_{ID} = h_1(ID, T_{ID})$ and $d_{ID} = t_{ID} + s\alpha_{ID}(mod\ q)$. Then, the KGC sends $(d_{ID}, T_{ID})$ to U.

- **Set-Secret-Value**: Input U's identity $ID$. U chooses a random number $x_{ID} \in Z_q^*$ as its secret value.

- **Set-Private-Key**: This sets U's private key as $SK_{ID} = (x_{ID}, d_{ID})$.

- **Set-Public-Key**: U computes $P_{ID_S} = x_{ID_S}P$ and sets $PK_{ID} = (P_{ID}, T_{ID})$ as its public key.

- **CLPAEKS**: Let $W = \{w_i | i = 1, 2, \cdots, m\}$ be a set of keywords. Take $prms$, $ID$, $PK_{ID}$ as input. U encrypts the keyword $w$ as follows:

  1. Compute $\beta_{ID} = h_2(ID, P_{ID}, T_{ID})$, choose a random number $r_i \in Z_q^*$ and compute $U_i = r_i P$, $Q_i = H_3(w_i)$.

  2. Compute $\Gamma_i = e(r_i Q_i, \beta_{ID} P_{ID} + T_{ID} + \alpha_{ID} P_{pub})$ and $v_i = h_4(\Gamma_i)$.
     The final ciphertext for the keyword is $C = \{C_1, C_2, \cdots, C_m\}$, where $C_i = (U_i, v_i)$.

- **Trapdoor**: This takes $prms$, $ID$, $SK_{ID}$, $PK_{ID}$ as input. U runs the following steps to compute the trapdoor $T_w$:

  1. Compute $\beta_{ID} = h_2(ID, P_{ID}, T_{ID})$ and $Q = H_3(w)$.

  2. Compute $T_w = (d_{ID} + \beta_{ID} x_{ID})Q$.

- **Test**: Take $prms$, the trapdoor $T_w$ and ciphertext $C$ as input. The cloud server checks whether $h_4(e(T_w, U_i)) = v_i$ holds. If it holds, then the server outputs 1. Otherwise, it outputs 0.

**Security vulnerability.**  In this subsection, we show that the scheme is vulnerable to an off-line keyword guessing attack. We prove that a malicious adversary can retrieve keyword-specific information from any query message captured by the protocol.

**Lemma 1** *Ma et al.'s scheme is susceptible to an off-line keyword guessing attack.*
**Proof 1** *An attacker $\mathcal{B}$ performs the following steps.*

1. $\mathcal{B}$ *first captures a valid trapdoor $T_w$. The goal of $\mathcal{B}$ is to recover $w$ from $T_w$. $\mathcal{B}$ guesses an appropriate keyword $w'$, and computes $H_3(w')$, $\beta_{ID} = h_2(ID, P_{ID}, T_{ID})$ and $\alpha_{ID} = h_1(ID, T_{ID})$.*

2. $\mathcal{B}$ *checks whether $e(T_w, P) = e(\beta_{ID} H_3(w'), P_{ID}) \cdot e(H_3(w'), T_{ID} + \alpha_{ID} P_{pub})$. If the equation holds, the guessed keyword is a valid keyword, namely, $w' = w$. Otherwise, go to Step (1). Specifically, if $w' = w$, then*

$$
\begin{aligned}
e(T_w, P) &= e((\beta_{ID} x_{ID} + d_{ID})Q, P) \\
&= e((\beta_{ID} x_{ID} + d_{ID})H_3(w), P) \\
&= e(\beta_{ID} x_{ID} H_3(w), P) \cdot e(d_{ID} H_3(w), P) \\
&= e(\beta_{ID} H_3(w), P_{ID}) \cdot e(H_3(w), d_{ID} P) \\
&= e(\beta_{ID} H_3(w), P_{ID}) \cdot e(H_3(w), (t_{ID} + s\alpha_{ID})P) \\
&= e(\beta_{ID} H_3(w'), P_{ID}) \cdot e(H_3(w'), T_{ID} + \alpha_{ID} P_{pub})
\end{aligned}
$$

## Definitions and system model

### System model

We first describe the relationships and interactions among the four entities, namely, the cloud server, KGC, data sender and data receiver, in dCLPAEKS: The KGC generates the system parameters and part of the user's private key according to the user's identity. The sender extracts keywords from each data file and uses the sender's secret key $Sk_{ID_s}$, the receiver's public key $Pk_{ID_r}$ and the server's public key $Pk_{Csvr}$ to encrypt keywords to form the dCLPAEKS ciphertext; then, it encrypts the file by using another encryption algorithm and sends it to the cloud server along with the keyword ciphertext. To search encrypted files, the receiver uses his secret key $Sk_{ID_r}$ and the sender's public key $Pk_{ID_s}$ to generate the trapdoor $T_w$ of the keywords and sends it to the cloud server. The cloud server uses its secret key to search and return the ciphertext files containing the target keywords.

### Definition of dCLPAEKS

Our dCLPAEKS scheme consists of the following (probabilistic) polynomial-time algorithms.

- **Setup ($\lambda$):** Given a security parameter $\lambda$, this algorithm generates a master public/secret key pair ($m_{pk}$, $m_{sk}$) and global parameter params.

- **KGen$_{Csvr}$ (params):** Given params, it generates a public/secret key pair ($PK_{Csvr}$, $SK_{Csvr}$) for the cloud server.

- **PPKGen (params, ID, $m_{sk}$):** Given params, a master secret key $m_{sk}$ and a user's identity $ID$, it generates the user's partial private key, referred to as $PPK_{ID}$.

- **SVGen (params, ID):** Given params and a user's identity $ID$, it generates a secret value, referred to as $SV_{ID}$.

- **SKGen (params, $PPK_{ID}$, $SV_{ID}$):** Given params, a partial private key $PPK_{ID}$ and a secret value $SV_{ID}$, it generates the user's secret key, referred to as $SK_{ID}$.

- **PKGen (params, $SV_{ID}$):** Given params and a secret value $SV_{ID}$, it generates a public key $PK_{ID}$ for the identity $ID$.

- **PEKS (params, $w$, $PK_{Csvr}$, $SK_{ID_s}$, $PK_{ID_r}$, $ID_s$, $ID_r$)**: Given params, a keyword $w$, $PK_{Csvr}$, a sender's identity $ID_s$ and $SK_{ID_s}$, and a receiver's identity $ID_r$ and $PK_{ID_r}$, it generates a cipher-text $C_w$.

- **Trapdoor (params, $w$, $SK_{ID_r}$, $PK_{ID_s}$, $ID_r$, $ID_s$)**: Given params, a keyword $w$, a receiver's identity $ID_r$ and $SK_{ID_r}$, and a sender's identity $ID_s$ and $PK_{ID_s}$, it generates a trapdoor $T_w$.

- **dTest (params, $SK_{Csvr}$, $C_w$, $T_w$, $ID_s$, $ID_r$)**: Given params, the server's secret key $SK_{Csvr}$, a PEKS ciphertext $C_w$, a trapdoor $T_w$, the identity $ID_s$ of a sender and the $ID_r$ of a receiver, it outputs 1 if $C_w$ and $T_w$ contain the same keyword, and 0 otherwise.

## Security models

There are two types of adversaries, i.e., a Type 1 adversary $\mathcal{A}_I$ and a Type 2 adversary $\mathcal{A}_{II}$, in certificateless cryptography [43]. Adversary $\mathcal{A}_I$ cannot access the master key. However, $\mathcal{A}_I$ can extract partial private keys and secret keys, request public keys and replace public keys with any values he chooses. Adversary $\mathcal{A}_{II}$ can access the system's master key, but cannot replace the user's public key.

We define the following five games between the adversaries $\mathcal{A}_I$ ($\mathcal{A}_{II}$) and a challenger $\mathcal{B}$ to show that our scheme is semantically secure against IKGA.

For adversaries in Game 1 to Game 4, we set the following natural restrictions.

1. The adversary cannot extract the secret key for the challenge identities.

2. The adversary cannot make a ciphertext query and trapdoor query on the challenge keywords $w_0^*$, $w_1^*$ for the challenge identity $ID_s^*$ of a sender and $ID_r^*$ of a receiver.

   **Ciphertext indistinguishability.** Game 1: Ciphertext indistinguishability for $\mathcal{A}_I$

   In this game, we set the semi-trusted cloud server as the adversary $\mathcal{A}_I$. Ciphertext indistin-guishability ensures that the ciphertext reveals no information about the underlying keyword to the cloud server.

- **Setup**: Given a security parameter $\lambda$, the challenger $\mathcal{B}$ generates the system parameter params, the PKG's public/secret key ($m_{pk}$, $m_{sk}$), and the server's public/secret key ($PK_{Csvr}$, $SK_{Csvr}$). It then invokes $\mathcal{A}_I$ on the input params and ($PK_{Csvr}$, $SK_{Csvr}$).

- **Phase 1**: Adversary $\mathcal{A}_I$ issues a sequence of queries adaptively polynomial-many times but is subject to the restrictions defined above.

  - *Partial Private Key Extraction*: Given the user's identity $ID$, it returns the user's partial pri-vate key $PPK_{ID}$ to $\mathcal{A}_I$.

  - *Secret Key Queries*: Given the user's identity $ID$, $\mathcal{B}$ returns the user's secret key $SK_{ID}$ to $\mathcal{A}_I$.

  - *Public Key Queries*: Given the user's identity $ID$, $\mathcal{B}$ returns the user's public key $PK_{ID}$ to $\mathcal{A}_I$.

  - *Replace Public Key*: $\mathcal{A}_I$ can replace the public key with any value he chooses.

  - *Ciphertext Queries*: Given a keyword $w$, identity $ID_s$ of a sender and identity $ID_r$ of a receiver, $\mathcal{B}$ computes the corresponding ciphertext $C_w$ and returns it to $\mathcal{A}_I$.

  - *Trapdoor Queries*: Given a keyword $w$, identity $ID_s$ of a sender and identity $ID_r$ of a receiver, $\mathcal{B}$ computes the corresponding trapdoor $T_w$ and returns it to $\mathcal{A}_I$.

- **Challenge**: $\mathcal{A}_I$ outputs two keywords $w_0^*, w_1^*$, the challenge identity $ID_s^*$ of a sender and $ID_r^*$ of a receiver, and $\mathcal{B}$ randomly chooses a bit $b \in \{0, 1\}$, computes the challenge ciphertext $C_{w_b^*}$ and returns it to $\mathcal{A}_I$, where $C_{w_b^*} = PEKS(params, w_b^*, PK_{Csvr}, SK_{ID_s^*}, PK_{ID_r^*}, ID_s^*, ID_r^*)$.

- **Phase 2**: Adversary $\mathcal{A}_I$ continues to issue requests to $\mathcal{B}$, as in phase 1.

- **Guess**: $\mathcal{A}_I$ outputs a bit $b' \in \{0, 1\}$, and wins the game if and only if $b' = b$.
  The advantage of $\mathcal{A}_I$ winning Game 1 is defined as

$$Adv_{\mathcal{A}_I}^C = |\mathrm{Pr}_{[b'=b]} - \frac{1}{2}|.$$

Game 2: Ciphertext indistinguishability for $\mathcal{A}_{II}$
In this game, we set the semi-trusted KGC as the adversary $\mathcal{A}_{II}$.

- **Setup**: $\mathcal{B}$ generates the system public parameter params, the PKG's public/secret key ($m_{pk}$, $m_{sk}$), and the server's public/secret key ($PK_{Csvr}$, $SK_{Csvr}$). Then, $\mathcal{B}$ returns params and $m_{sk}$ to $\mathcal{A}_{II}$.

- **Phase 1**: $\mathcal{A}_{II}$ can adaptively issue a sequence of queries polynomial-many times but obeys the restrictions defined above.

  - *Secret Key Queries*: Taking the identity $ID$ as input, $\mathcal{B}$ outputs the secret key $SK_{ID}$ to $\mathcal{A}_{II}$.

  - *Public Key Queries*: Taking the identity $ID$ as input, $\mathcal{B}$ outputs the public key $PK_{ID}$ to $\mathcal{A}_{II}$.

  - *Ciphertext Queries*: Given a keyword $w$, identity $ID_s$ of a sender and $ID_r$ of a receiver, $\mathcal{B}$ outputs the corresponding ciphertext $C_w$ to $\mathcal{A}_{II}$.

  - *Trapdoor Queries*: Given a keyword $w$, identity $ID_s$ of a sender and $ID_r$ of a receiver, $\mathcal{B}$ outputs the corresponding trapdoor $T_w$ to $\mathcal{A}_{II}$.

- **Challenge**: $\mathcal{A}_{II}$ outputs two keywords $w_0^*$ and $w_1^*$ and the challenge identity $ID_s^*$ of a sender and $ID_r^*$ of a receiver, and $\mathcal{B}$ randomly chooses a bit $b \in \{0, 1\}$, computes the challenge ciphertext $C_{w_b^*}$ and returns it to $\mathcal{A}_{II}$, where
  $C_{w_b^*} = PEKS(params, w_b^*, PK_{Csvr}, SK_{ID_s^*}, PK_{ID_r^*}, ID_s^*, ID_r^*)$.

- **Phase 2**: Adversary $\mathcal{A}_{II}$ continues to issue queries to $\mathcal{B}$ as in phase 1.

- **Guess**: $\mathcal{A}_{II}$ outputs a bit $b' \in \{0, 1\}$, and wins the game if and only if $b' = b$.
  The advantage of $\mathcal{A}_{II}$ winning Game 2 is defined as

$$Adv_{\mathcal{A}_{II}}^C(\lambda) = |\mathrm{Pr}_{[b'=b]} - \frac{1}{2}|.$$

**Definition 3** *We say that a* dCLPAEKS *scheme satisfies ciphertext indistinguishability if, for any polynomial-time adversaries* $\mathcal{A}_i(i = I, II)$, $Adv_{\mathcal{A}_i}^C(\lambda)$ *is negligible.*

**Trapdoor indistinguishability.** Game 3: Trapdoor indistinguishability for $\mathcal{A}_I$
Similar to Game 1, we set the semi-trusted cloud server as the adversary $\mathcal{A}_I$. Trapdoor indistinguishability guarantees that the cloud server cannot obtain any information about the keyword from a given trapdoor. This indicates that the server cannot forge valid ciphertext and cannot perform offline inside keyword guessing attacks (IKGAs) successfully.

- **Setup**: Same as in Game 1.

- **Phase 1**: Same as in Game 1.

- **Challenge**: $\mathcal{A}_I$ outputs two keywords $w_0^*$ and $w_1^*$ and the challenge identity $ID_s^*$ of a sender and $ID_r^*$ of a receiver, and $\mathcal{B}$ randomly chooses a bit $b \in \{0, 1\}$, computes the challenge trapdoor $T_{w_b^*}$ and returns it to $\mathcal{A}_I$, where $T_{w_b^*} = Trapdoor(params, w_b^*, SK_{ID_r^*}, PK_{ID_s^*}, ID_s^*, ID_r^*)$.

- **Phase 2**: Adversary $\mathcal{A}_I$ continues to issue queries to $\mathcal{B}$, as in phase 1.

- **Guess**: $\mathcal{A}_I$ outputs a bit $b' \in \{0, 1\}$, if $b' = b$, we say $\mathcal{A}_I$ wins the game. The advantage of $\mathcal{A}_I$ in breaking trapdoor indistinguishability in a dCLPAEKS is defined as

$$Adv_{\mathcal{A}_I}^T = |\Pr_{[b'=b]} - \frac{1}{2}|.$$

Game 4: Trapdoor indistinguishability for $\mathcal{A}_{II}$

Game 4 is similar to Game 2; the difference is that $\mathcal{B}$ needs to generate a trapdoor of the challenge keywords for the challenge identity $ID_s^*$ of a sender and $ID_r^*$ of a receiver in Game 4.

**Definition 4** *We say that a* dCLPAEKS *scheme satisfies trapdoor indistinguishability if for any polynomial-time adversaries* $\mathcal{A}_i (i = I, II)$, $Adv_{\mathcal{A}_i}^T(\lambda)$ *is negligible.*

**Designated testability.** Game 5: In this game, we assume that $\mathcal{A}_I$ is an outside adversary who is allowed to obtain any user's secret key. Designated testability aims to prevent adversaries from searching the ciphertexts while guaranteeing that only the designated server can.

- **Setup**: $\mathcal{B}$ generates the system public parameter params, the PKG's public/secret key ($m_{pk}$, $m_{sk}$), and the server's public/secret key ($PK_{Csvr}$, $SK_{Csvr}$). It invokes $\mathcal{A}_I$ on input params and $PK_{Csvr}$.

- **Phase1**: Adversary $\mathcal{A}_I$ can adaptively issue a sequence of queries polynomial-many times.

  - *Secret Key Queries*: Given a user's identity $ID$, $\mathcal{B}$ return the user's secret key $SK_{ID}$ to $\mathcal{A}_I$.

  - *Public Key Queries*: Given a user's identity $ID$, $\mathcal{B}$ return the user's public key $PK_{ID}$ to $\mathcal{A}_I$.

- **Challenge**: $\mathcal{A}_I$ outputs two keywords $w_0^*$ and $w_1^*$ and the challenge identity $ID_s^*$ of a sender and $ID_r^*$ of a receiver, and $\mathcal{B}$ randomly chooses a bit $b \in \{0, 1\}$, computes the challenge ciphertext $C_{w_b^*}$ and returns it to $\mathcal{A}_I$, where
  $C_{w_b^*} = PEKS(params, w_b^*, PK_{Csvr}, SK_{ID_s^*}, PK_{ID_r^*}, ID_s^*, ID_r^*)$.

- **Phase 2**: Adversary $\mathcal{A}_I$ continues to issue queries to $\mathcal{B}$ as in phase 1.

- **Guess**: $\mathcal{A}_I$ outputs a bit $b' \in \{0, 1\}$, and if $b' = b$, $\mathcal{A}_I$ wins the game.
  The advantage of $\mathcal{A}_I$ winning Game 5 is defined as

$$Adv_{\mathcal{A}_I}^D = |\Pr_{[b'=b]} - \frac{1}{2}|.$$

**Definition 5** *We say that a* dCLPAEKS *scheme satisfies designated testability if for any polynomial-time adversary* $\mathcal{A}_I$, $Adv_{\mathcal{A}_I}^D(\lambda)$ *is negligible.*

## A instantiation of dCLPAEKS

### Concrete dCLPAEKS scheme

Here, we present a concrete dCLPAEKS scheme, which is composed of nine polynomial-time algorithms.

- **Setup ($\lambda$)**: Given a security parameter $\lambda$, this algorithm runs as follows:

  1. Select two cyclic groups $\mathbb{G}_1$ and $\mathbb{G}_2$ with the same prime order $p$, a bilinear pairing $\hat{e} : \mathbb{G}_1 \times \mathbb{G}_1 \to \mathbb{G}_2$, and a cryptographic hash function $H_1 : \{0,1\}^* \to \mathbb{G}_1$.

  2. Select a random number $\alpha \in Z_p^*$ as the master key $m_{sk}$ and set $m_{pk} = g^\alpha$, where $g$ is an arbitrary generator of $\mathbb{G}_1$.

  3. Choose another arbitrary generator $h \in \mathbb{G}_1$.

  4. Choose two additional cryptographic hash functions $H : \mathbb{G}_2 \times \{0,1\}^* \to \mathbb{G}_1$ and $H_2 : \{0,1\}^* \times \{0,1\}^* \to \mathbb{G}_1$.
     The system parameters params $= (\mathbb{G}_1, \mathbb{G}_2, \hat{e}, g, h, H, H_1, H_2, m_{pk})$ are publicly and authentically available, but only the KGC knows the master key $m_{sk}$. Steps (1) and (2) of the algorithm are run by the KGC.

- **KGen$_{Csvr}$ (params)**: Chooses $v \in Z_p^*$ randomly, and outputs the server's public/secret key pair $(PK_{Csvr}, SK_{Csvr}) = (g^v, v)$.

- **PPKGen (params, $ID_i$, $m_{sk}$)**: Outputs the partial private key $PPK_{ID_i} = H_1(ID_i)^\alpha$.

- **SVGen (params, $ID_i$)**: Selects $\beta_{ID_i} \in Z_p^*$ randomly, and outputs the secret value $SV_{ID_i} = \beta_{ID_i}$.

- **SKGen (params, $PPK_{ID_i}$, $SV_{ID_i}$)**: Outputs the secret key $SK_{ID_i} = (SK_{ID_i}^1, SK_{ID_i}^2) = (H_1(ID_i)^\alpha, \beta_{ID_i})$.

- **PKGen (params, $SV_{ID}$)**: Outputs the public key $PK_{ID_i} = g^{\alpha\beta_{ID_i}}$.

- **PEKS (params, $w$, $PK_{Csvr}$, $SK_{ID_s}$, $PK_{ID_r}$, $ID_s$, $ID_r$)**: Selects $s \in Z_p^*$ randomly and computes $C_1 = \hat{e}(H(k,w), PK_{Csvr})^s$, $C_2 = g^s$, $C_3 = h^s$, where $k = \hat{e}(SK_{ID_s}^1, H_1(ID_r)) \cdot \hat{e}(H_2(ID_s, ID_r), PK_{ID_r})^{SK_{ID_s}^2}$ outputs the ciphertext $C_w = (C_1, C_2, C_3)$.

- **Trapdoor (params, $w$, $SK_{ID_r}$, $PK_{ID_s}$, $ID_r$, $ID_s$)**: Selects $r \in Z_p^*$ randomly, and computes $T_1 = H(k,w) \cdot h^r$, $T_2 = g^r$, where $k = \hat{e}(H_1(ID_s), SK_{ID_r}^1) \cdot \hat{e}(H_2(ID_s, ID_r), PK_{ID_s})^{SK_{ID_r}^2}$, and outputs the trapdoor $T_w = (T_1, T_2)$.

- **dTest (params, $SK_{Csvr}$, $C_w$, $T_w$, $ID_s$, $ID_r$)**: Returns 1 if $C_1 \cdot \hat{e}(T_2^{SK_{Csvr}}, C_3) = \hat{e}(T_1^{SK_{Csvr}}, C_2)$ and 0 otherwise.

### Security analysis

In this subsection, we analyze the security of the above concrete construction.

dCLPAEKS **ciphertext indistinguishability.** **Theorem 1** *Our* dCLPAEKS *scheme satisfies ciphertext indistinguishability under the assumption that DBDH is intractable.*

This conclusion is derived from the following two lemmas.

**Lemma 2** *For any polynomial-time adversary $\mathcal{A}_{\mathcal{I}}$, our dCLPAEKS scheme satisfies ciphertext indistinguishability in Game 1 under the random oracle model assuming DBDH is intractable.*

**Proof 2** *Assume that $\mathcal{A}_{\mathcal{I}}$ is a semi-trusted server that tries to break the ciphertext indistinguishability of our dCLPAEKS scheme. We construct a simulator $\mathcal{B}$ to solve the DBDH problem. Given a random challenge $(\mathbb{G}_1, \mathbb{G}_2, \hat{e}, g, g^x, g^y, g^z, Z)$, where $Z$ is either equal to $\hat{e}(g,g)^{xyz}$ or a random element of $\mathbb{G}_2$, $\mathcal{B}$ interacts with $\mathcal{A}_{\mathcal{I}}$ as follows:*

- **Setup**: *$\mathcal{B}$ randomly chooses $h$ from $\mathbb{G}_1$ and $v \in Z_p^*$ and sets $(PK_{Csvr}, SK_{Csvr}) = (g^v, v)$ and $params = (\mathbb{G}_1, \mathbb{G}_2, \hat{e}, p, g, h, m_{pk} = g^z)$. $\mathcal{B}$ sends params and $(PK_{Csvr}, SK_{Csvr})$ to $\mathcal{A}_{\mathcal{I}}$.*

- **Phase1**: *$\mathcal{A}_{\mathcal{I}}$ is allowed to issue queries to the following oracles simulated by $\mathcal{B}$. To simplify, let us make the following assumptions: the adversary does not initiate repeated queries, and the attacker does not use an identity to perform any calculations before initiating $H_1$ on the identity.*

  - *H Queries: Upon receiving $\mathcal{A}_{\mathcal{I}}$s query on an element $k$ and a keyword $w$, $\mathcal{B}$ randomly chooses an element from $\mathbb{G}_1$ as the output of $H(k, w)$.*

  - *$H_1$ Queries: Suppose that $\mathcal{A}_{\mathcal{I}}$ makes at most $q_{H_1}$ queries. A list is maintained by $\mathcal{B}$, referred to as $L_{H_1}$. $\mathcal{B}$ randomly chooses $i, j \in \{1, 2, \cdots, q_{H_1}\}$, and guesses that the i-th and the j-th $H_1$ queries initiated by $\mathcal{A}_{\mathcal{I}}$ correspond to the sender's challenge identity $ID_s^*$ and receiver's challenge identity $ID_r^*$, respectively. When $\mathcal{A}_{\mathcal{I}}$ makes a $H_1$ query on identity ID, $\mathcal{B}$ responds as follows:*

    1. *If this is the i-th query, e.g., $ID = ID_s^*$, $\mathcal{B}$ outputs $H_1(ID) = g^x$ and adds $< ID, g^x, \bot >$ to $L_{H_1}$.*

    2. *If this is the j-th query, e.g., $ID = ID_r^*$, $\mathcal{B}$ outputs $H_1(ID) = g^y$ and adds $< ID, g^y, \bot >$ to $L_{H_1}$.*

    3. *Otherwise, $\mathcal{B}$ chooses a random number $\mu_{ID} \in Z_p^*$, outputs $H_1(ID) = g^{\mu_{ID}}$ and adds $< ID, H_1(ID), \mu_{ID} >$ to $L_{H_1}$.*

  - *$H_2$ Queries: Given a pair of identities $(ID_s, ID_r)$, $\mathcal{B}$ randomly chooses an element from $\mathbb{G}_1$ as the output of $H_2(ID_s, ID_r)$.*

  - *Partial Private Key Extraction: When $\mathcal{A}_{\mathcal{I}}$ asks for the partial private key of the ID, if $ID = ID_s^*$ or $ID = ID_r^*$, $\mathcal{B}$ outputs a random bit $\eta'$ and aborts. Otherwise, it recovers the tuple $< ID, H_1(ID), \mu_{ID} >$ from $L_{H_1}$, and returns the partial private key $PPK_{ID} = (g^z)^{\mu_{ID}}$ to $\mathcal{A}_{\mathcal{I}}$.*

  - *Secret Key Queries: $\mathcal{B}$ maintains a list $L_{SK}$, which is initially empty. Taking ID as input, $\mathcal{B}$ performs the following actions:*

    1. *If $ID \neq ID_s^*$ and $ID \neq ID_r^*$, it recovers the tuple $< ID, H_1(ID), \mu_{ID} >$ from $L_{H_1}$ and chooses a random number $\beta_{ID}$ as the secret value. Then, $\mathcal{B}$ returns the secret key $SK_{ID} = ((g^z)^{\mu_{ID}}, \beta_{ID})$ to $\mathcal{A}_{\mathcal{I}}$ and adds $< ID, SK_{ID} >$ into $L_{SK}$.*

    2. *Otherwise, $\mathcal{B}$ randomly chooses an element $\beta_{ID} \in Z_p^*$ as a secret value and adds $< ID, \bot, \beta_{ID} >$ into $L_{SK}$. Then, $\mathcal{B}$ outputs a random bit $\eta'$ and aborts.*

  - *Public Key Queries: $\mathcal{B}$ maintains a list $L_{PK}$, which is initially empty. Given an identity ID, $\mathcal{B}$ retrieves the tuple $< ID, SK_{ID} >$ from $L_{SK}$, computes the public key $PK_{ID} = (g^z)^{\beta_{ID}}$, and then returns it to $\mathcal{A}_{\mathcal{I}}$ and adds $< ID, PK_{ID} >$ into $L_{PK}$.*

- *Replace Public Key*: $\mathcal{A}_\mathcal{I}$ can replace the public key with any value he chooses.

- *Ciphertext Queries*: *Taking $(w, ID_r, ID_s)$ as input, $\mathcal{B}$ randomly chooses $s \in Z_p^*$ and executes the following steps:*

  1. *If at least one of $ID_r$ and $ID_s$ is not equal to $ID_r^*$ or $ID_s^*$, without loss of generality, we assume that $ID_s \notin \{ID_s^*, ID_r^*\}$. $\mathcal{B}$ recovers $< ID_s, H_1(ID_s), \mu_{ID_s} >$ from $L_{H_1}$, $< ID_s, SK_{ID_s} >$ from $L_{SK}$, and $< ID_r, PK_{ID_r} >$ from $L_{PK}$, computes $k = \hat{e}(g^z, H_1(ID_r))^{\mu_{ID_s}} \cdot \hat{e}(H_2(ID_s, ID_r), PK_{ID_r})^{SK_{ID_s}^2}$ and returns $C_1 = \hat{e}(H(k, w), PK_{Csvr})^s$, $C_2 = g^s$, $C_3 = h^s$.*

  2. *Otherwise, $\mathcal{B}$ outputs a random bit $\eta'$ and aborts.*

- *Trapdoor Queries*: *Taking $(w, ID_r, ID_s)$ as input, $\mathcal{B}$ randomly chooses $r \in Z_p^*$ and responds as follows:*

  1. *If at least one of $ID_r$ and $ID_s$ is not equal to $ID_r^*$ or $ID_s^*$, without loss of generality, we assume that $ID_s \notin \{ID_s^*, ID_r^*\}$. $\mathcal{B}$ recovers $< ID_s, H_1(ID_s), \mu_{ID_s} >$ from $L_{H_1}$, $< ID_r, SK_{ID_r} >$ from $L_{SK}$, and $< ID_s, PK_{ID_s} >$ from $L_{PK}$, computes $k = \hat{e}(g^z, H_1(ID_r))^{\mu_{ID_s}} \cdot \hat{e}(H_2(ID_s, ID_r), PK_{ID_s})^{SK_{ID_r}^2}$ and returns $T_1 = H(k, w) \cdot h^r$, $T_2 = g^r$.*

  2. *Otherwise, $\mathcal{B}$ outputs a random bit $\eta'$ and aborts.*

- **Challenge**: *$\mathcal{A}_\mathcal{I}$ issues a challenge on two different keywords $w_0^*$, $w_1^*$, a sender's identity $ID_s^*$ and a receiver's identity $ID_r^*$. $\mathcal{B}$ randomly selects a bit $b \in \{0, 1\}$ and an element $s \in Z_p^*$, and computes the ciphertext $C_{w_b^*} = (C_1^*, C_2^*, C_3^*)$, where $C_1^* = \hat{e}(H(Z \cdot \hat{e}(H_2(ID_s^*, ID_r^*), PK_{ID_r^*})^{SK_{ID_s^*}^2}, w_b^*), PK_{Csvr})^s$ and $C_2^* = g^s$, $C_3^* = h^s$.*

- **Phase 2**: *Simulator $\mathcal{B}$ responds as in phase 1.*

- **Guess**: *$\mathcal{A}_\mathcal{I}$ outputs a bit $b' \in \{0, 1\}$, and then $\mathcal{B}$ outputs $\eta' = 0$ if $b' = b$ and 1 otherwise.*

*If $\mathcal{B}$ guesses that the challenge identities are incorrect, $\mathcal{B}$ aborts. Denote by abt the event that $\mathcal{B}$ aborts. The probability that event abt does not occur is $1/q_{H_1} \cdot (q_{H_1} - 1)$.*

*Assume that $\mathcal{B}$ does not abort in the game. If $Z = \hat{e}(g, g)^{xyz}$, the view of $\mathcal{A}_\mathcal{I}$ is the same as in a real attack, and $\mathcal{A}_\mathcal{I}$ succeeds in the game with probability $Adv_{\mathcal{A}_\mathcal{I}}^C = \|\Pr_{[b'=b]} - \frac{1}{2}\|$. If $Z$ is selected from $\mathbb{G}_2$ randomly, then $k$ is also a random element in $\mathbb{G}_1$, so $\mathcal{A}_\mathcal{I}$ wins Game 1 with probability at most $\frac{1}{2}$. Hence, the advantage of $\mathcal{B}$ in solving the DBDH problem is*

$$Adv_\mathcal{B}^{DBDH}$$
$$= |\Pr_{[\eta'=\eta|abt]} \cdot \Pr_{[abt]} + \Pr_{[\eta'=\eta|\neg abt]} \cdot \Pr_{[\neg abt]} - \frac{1}{2}|$$
$$= |\frac{1}{2}(1 - \Pr_{[\neg abt]}) + (\Pr_{[\eta'=0|\neg abt \wedge \eta=0]} \cdot \Pr_{[\eta=0]} + \Pr_{[\eta'=1|\neg abt \wedge \eta=1]} \cdot \Pr_{[\eta=1]}) \cdot \Pr_{[\neg abt]} - \frac{1}{2}|$$
$$\geq |\frac{1}{2}(1 - \Pr_{[\neg abt]}) + \Pr_{[\neg abt]} \cdot \left(\frac{1}{2}\left(Adv_{\mathcal{A}_\mathcal{I}}^C(\lambda) + \frac{1}{2}\right) + \frac{1}{2} \cdot \frac{1}{2}\right) - \frac{1}{2}|$$
$$= \frac{1}{2}\Pr_{[\neg abt]} \cdot Adv_{\mathcal{A}_\mathcal{I}}^C(\lambda)$$
$$= \frac{1}{2q_{H_1} \cdot (q_{H_1} - 1)} \cdot Adv_{\mathcal{A}_\mathcal{I}}^C(\lambda)$$

*If $Adv_{\mathcal{A}_I}^C(\lambda)$ is not negligible, then $Adv_\mathcal{B}^{DBDH}$ is not negligible.*

**Lemma 3** *For any polynomial-time adversary $\mathcal{A}_{\mathcal{II}}$, our dCLPAEKS scheme satisfies ciphertext indistinguishability in Game 2 under the random oracle model, assuming DBDH is intractable.*

**Proof 3** *Assume that $\mathcal{A}_{\mathcal{II}}$ is a semi-trusted KGC that tries to break the ciphertext indistinguishability of our dCLPAEKS scheme. Given a DBDH instance $(\mathbb{G}_1, \mathbb{G}_2, \hat{e}, p, g, g^x, g^y, g^z, Z)$, we will construct an algorithm $\mathcal{B}$ to solve the DBDH problem by using $\mathcal{A}_{\mathcal{II}}$ as a subroutine. $\mathcal{B}$ interacts with $\mathcal{A}_{\mathcal{II}}$ as follows:*

- **Setup**: *$\mathcal{B}$ selects $h$ from $\mathbb{G}_1$ and $\alpha \in Z_p^*$ randomly and sets params $= (\mathbb{G}_1, \mathbb{G}_2, \hat{e}, p, g, h, m_{pk} = g^\alpha)$ and $PK_{Csvr} = g^x$. Then, $\mathcal{B}$ sends params, $PK_{Csvr}$ and $m_{sk} = \alpha$ to $\mathcal{A}_{\mathcal{II}}$.*

- **Phase 1**: *$\mathcal{A}_{\mathcal{II}}$ executes the following queries; assume that $\mathcal{A}_{\mathcal{II}}$ does not repeat its queries.*

  - *H Queries: A list is maintained by $\mathcal{B}$, referred to as $L_H$, which is initially empty. Taking an element $k$ and a keyword $w$ as input, $\mathcal{B}$ randomly chooses $\mu_{k,w} \in Z_p^*$, returns $H(k, w) = g^y g^{\mu_{k,w}}$ to $\mathcal{A}_{\mathcal{II}}$ and adds $< (k, w), H(k, w), \mu_{k,w} >$ into $L_H$.*

  - *$H_1$ Queries: Given an identity ID, $\mathcal{B}$ randomly selects an element from $\mathbb{G}_1$ as the $H_1(ID)$ value and returns it to $\mathcal{A}_{\mathcal{II}}$.*

  - *$H_2$ Queries: Given a pair of identities $(ID_s, ID_r)$, $\mathcal{B}$ randomly chooses an element from $\mathbb{G}_1$ as its $H_2(ID_s, ID_r)$ value, and outputs it to $\mathcal{A}_{\mathcal{II}}$.*

  - *Secret Key Queries: $\mathcal{B}$ maintains a list $L_{SK}$ that is initially empty. Taking an identity ID as input, $\mathcal{B}$ selects a random number $\beta_{ID} \in Z_p^*$ as the secret value and returns the secret key $SK_{ID} = ((H_1(ID)^\alpha, \beta_{ID})$ to $\mathcal{A}_{\mathcal{II}}$. Then, $\mathcal{B}$ adds $< ID, H_1(ID)^\alpha, \beta_{ID} >$ into $L_{SK}$.*

  - *Public Key Queries: $\mathcal{B}$ maintains a list $L_{PK}$ that is initially empty. Given an identity ID, $\mathcal{B}$ recovers the tuple $< ID, H_1(ID)^\alpha, \beta_{ID} >$ from $L_{SK}$, computes the public key $PK_{ID} = (g^\alpha)^{\beta_{ID}}$, and returns it to $\mathcal{A}_{\mathcal{II}}$. Then, $< ID, PK_{ID} >$ is added to $L_{PK}$.*

  - *Ciphertext Queries: Taking $(w, ID_r, ID_s)$ as input, $\mathcal{B}$ recovers the tuple $< (k, w), H(k, w), \mu_{k,w} >$ from $L_H$, where $k = \hat{e}(SK_{ID_s}^1, H_1(ID_r)) \cdot \hat{e}(H_2(ID_s, ID_r), PK_{ID_r})^{SK_{ID_s}^2}$. If there is no such tuple, $\mathcal{B}$ generates it as in previous queries. Then, $\mathcal{B}$ randomly chooses $s \in Z_p^*$ and computes the ciphertext $C_w = (C_1, C_2, C_3)$, where $C_1 = \hat{e}(H(k, w), PK_{Csvr})^s$ and $C_2 = g^s, C_3 = h^s$.*

  - *Trapdoor Queries: Taking $(w, ID_r, ID_s)$ as input, $\mathcal{B}$ recovers the tuple $< (k, w), H(k, w), \mu_{k,w} >$ from $L_H$, where $k = \hat{e}(H_1(ID_s), SK_{ID_r}^1) \cdot \hat{e}(H_2(ID_s, ID_r), PK_{ID_s})^{SK_{ID_r}^2}$. If there is no such tuple, $\mathcal{B}$ generates it as in previous queries. Then, $\mathcal{B}$ randomly chooses $r \in Z_p^*$ and computes trapdoor $T_w = (T_1, T_2)$, where $T_1 = H(k, w) \cdot h^r, T_2 = g^r$.*

- **Challenge**: *$\mathcal{A}_{\mathcal{II}}$ submits two challenge keywords $w_0^*, w_1^*$, the sender's challenge identity $ID_s^*$ and the receiver's challenge identity $ID_r^*$. $\mathcal{B}$ chooses a random bit $b \in \{0, 1\}$ and recovers the tuple $< (k^*, w_b^*), H((k^*, w_b^*), \mu_{k^*, w_b^*} >$, where $k^* = \hat{e}(SK_{ID_s^*}^1, H_1(ID_r^*)) \cdot \hat{e}(H_2(ID_s^*, ID_r^*), PK_{ID_r^*})^{SK_{ID_s^*}^2}$. If there is no such tuple, $\mathcal{B}$ generates it as in previous queries. Then, it computes the challenge ciphertext $C_{w_b^*} = (C_1^*, C_2^*, C_3^*)$, where $C_1^* = Z \cdot \hat{e}(g^z, g^x)^{\mu_{k^*, w_b^*}}, C_2^* = g^z, C_3^* = h^z$.*

- **Phase 2**: *Simulator $\mathcal{B}$ responds as in phase 1.*

- **Guess**: *$\mathcal{A}_{\mathcal{II}}$ outputs a bit $b'$; if $b' = b$, $\mathcal{B}$ outputs $\eta' = 0$, and it outputs 1 otherwise.*

*If $Z = \hat{e}(g, g)^{xyz}$, then the challenge ciphertext is a correctly distributed verifiable ciphertext, so the view of $\mathcal{A}_{\mathcal{II}}$ is the same as in real attack, and $\mathcal{A}_{\mathcal{II}}$ succeeds in Game 2 with probability*

$Adv^C_{\mathcal{A}_{II}} + \frac{1}{2}$. *If Z is selected from* $\mathbb{G}_2$ *randomly, then* $C^*_1$ *is also a random element in* $\mathbb{G}_1$; *hence,* $\mathcal{A}_{II}$ *succeeds in the game with probability at most* $\frac{1}{2}$. *Therefore, the advantage of* $\mathcal{B}$ *in solving the DBDH problem is*

$$Adv^{DBDH}_{\mathcal{B}}$$
$$= |\mathrm{Pr}_{[\eta'=\eta|\eta=1]} \cdot \mathrm{Pr}_{[\eta=1]} + \mathrm{Pr}_{[\eta'=\eta|\eta=0]} \cdot \mathrm{Pr}_{[\eta=0]} - \frac{1}{2}|$$
$$\geq |\frac{1}{2} \cdot \frac{1}{2} + \frac{1}{2}\left(Adv^C_{\mathcal{A}_{II}}(\lambda) + \frac{1}{2}\right) - \frac{1}{2}|$$
$$= \frac{1}{2} \cdot Adv^C_{\mathcal{A}_{II}}$$

*If* $\mathcal{A}^C_{II}(\lambda)$ *is not negligible, then* $Adv^{DBDH}_{\mathcal{B}}$ *is not negligible.*

**Trapdoor indistinguishability of** dCLPAEKS. **Theorem 2** *Our* dCLPAEKS *scheme satisfies trapdoor indistinguishability under the assumption that DBDH is intractable.*

This conclusion is derived from the following two lemmas.

**Lemma 4** *For any polynomial-time adversary* $\mathcal{A}_I$, *our* dCLPAEKS *scheme satisfies trapdoor indistinguishability in Game 3 under the random oracle model, assuming DBDH is intractable.*

The proof of Lemma 4 is similar to that of Lemma 2. The difference is that the simulator generates a challenge trapdoor in the challenge stage. We omit the proof details here.

**Lemma 5** *For any polynomial-time adversary* $\mathcal{A}_{II}$, *our* dCLPAEKS *scheme satisfies trapdoor indistinguishability in Game 4 under the random oracle model, assuming DBDH is intractable.*

**Proof 4** *Assume that* $\mathcal{A}_{II}$ *is a semi-trusted KGC that tries to break the trapdoor indistinguishability of our scheme. We construct a simulator* $\mathcal{B}$ *to solve the DBDH problem. Given a random challenge* $(\mathbb{G}_1, \mathbb{G}_2, \hat{e}, p, g, g^x, g^y, g^z, Z)$, $\mathcal{B}$ *interacts with* $\mathcal{A}_{II}$ *as follows*:

- **Setup**: $\mathcal{B}$ *selects h from* $\mathbb{G}_1$ *and* $v, \alpha \in Z^*_p$ *randomly and sets params* $=$
  $(\mathbb{G}_1, \mathbb{G}_2, \hat{e}, p, g, h, m_{pk} = g^\alpha)$ *and* $(PK_{Csvr}, SK_{Csvr}) = (g^v, v)$. $\mathcal{B}$ *returns params,* $PK_{Csvr}$ *and* $m_{sk}$ $= \alpha$ *to* $\mathcal{A}_{II}$.

- **Phase 1**: $\mathcal{B}$ *responds to* $\mathcal{A}_{II}$' *inquiry as follows, assuming that* $\mathcal{A}_{II}$ *does not initiate repeated queries.*

  - *H Queries: Given a keyword w and an element k,* $\mathcal{B}$ *randomly chooses an element from* $\mathbb{G}_1$ *as the output of H(k, w).*

  - $H_1$ *Queries: Given an identity ID,* $\mathcal{B}$ *picks a random number from* $\mathbb{G}_1$ *and returns it to* $\mathcal{A}_{II}$ *as the* $H_1(ID)$ *value of ID.*

  - $H_2$ *Queries: Suppose that* $\mathcal{A}_{II}$ *issues at most* $q_{H_2}$ *queries. A list is maintained by* $\mathcal{B}$, *referred to as* $L_{H_2}$, *which is initially empty.* $\mathcal{B}$ *randomly selects* $i \in \{1, 2, \cdots, q_{H_2}\}$ *and guesses that the two identities in the i-th inquiry are the sender's challenge identity* $ID^*_s$ *and receiver's challenge identity* $ID^*_r$, *respectively. Taking a pair of identities* $(ID_s, ID_r)$ *as input,* $\mathcal{B}$ *responds as follows:*

    1. *If this is the i-th query, e.g.,* $(ID_s, ID_r) = (ID^*_s, ID^*_r)$, *it returns* $H_2(ID_s, ID_r) = g^z$ *and adds* $< (ID_s, ID_r), g^z, \perp >$ *into* $L_{H_2}$.

    2. *Otherwise,* $\mathcal{B}$ *randomly chooses* $\mu_{r,s} \in Z^*_p$, *returns* $H_2(ID_s, ID_r) = g^{\mu_{r,s}}$ *and adds* $< (ID_s, ID_r), g^{\mu_{r,s}}, \mu_{r,s} >$ *into* $L_{H_2}$.

- *Secret Key Queries: A list is maintained by $\mathcal{B}$, called $L_{SK}$. Given an identity ID, $\mathcal{B}$ responds as follows:*

  1. *If $ID \notin (ID_s^*, ID_r^*)$, $\mathcal{B}$ selects a random number $\beta_{ID} \in Z_p^*$ as the secret value of the identity ID and returns the secret key $SK_{ID} = (H_1(ID)^a, \beta_{ID})$ to $\mathcal{A}_{II}$, then adds $< ID, SK_{ID} >$ into $L_{SK}$.*

  2. *Otherwise, $\mathcal{B}$ outputs a random bit $\eta'$ and aborts.*

- *Public Key Queries: $\mathcal{B}$ maintains an initially empty list $L_{PK}$. Taking an identity ID as input, $\mathcal{B}$ returns a value to $\mathcal{A}_{II}$ according to the following conditions:*

  1. *If $ID = ID_s^*$, it outputs $PK_{ID} = (g^x)^\alpha$ and adds $< ID, (g^x)^\alpha >$ into $L_{PK}$.*

  2. *If $ID = ID_r^*$, it outputs $PK_{ID} = (g^y)^\alpha$ and adds $< ID, (g^y)^\alpha >$ into $L_{PK}$.*

  3. *Otherwise, $\mathcal{B}$ retrieves the tuple $< ID, H_1(ID)^a, \beta_{ID} >$ from $L_{SK}$ and computes the public key $PK_{ID} = (g^\alpha)^{\beta_{ID}}$. Then, $\mathcal{B}$ returns $PK_{ID}$ to $\mathcal{A}_{II}$ and adds $< ID, PK_{ID} >$ into $L_{PK}$.*

- *Ciphertext Queries: Taking $(w, ID_r, ID_s)$ as input, $\mathcal{B}$ randomly chooses $s \in Z_p^*$ and computes the ciphertext $C_w = (C_1, C_2, C_3)$ as follows:*

  1. *If $(ID_r, ID_s) = (ID_r^*, ID_s^*)$ or $(ID_r, ID_s) = (ID_s^*, ID_r^*)$, $\mathcal{B}$ computes $C_1 = \hat{e}(H(\hat{e}(H_1(ID_s)^\alpha, H_1(ID_r)) \cdot Z^\alpha), w), PK_{Csvr})^s$ and $C_2 = g^s, C_3 = h^s$.*

  2. *Otherwise, at least one of $ID_r$ and $ID_s$ is not equal to $ID_r^*$ or $ID_s^*$. Without loss of generality, we assume that $ID_s \notin \{ID_s^*, ID_r^*\}$. $\mathcal{B}$ recovers $< (ID_s, ID_r), g^{\mu_{r,s}}, \mu_{r,s} >$ from $L_{H_2}$ and $< ID_s, PK_{ID_s} >$ from $L_{PK}$, computes $k = \hat{e}(H_1(ID_s)^\alpha, H_1(ID_r)) \cdot e(g^y, PK_{ID_s})^{\mu_{r,s}}$ and returns $C_1 = \hat{e}(H(k, w), PK_{Csvr})^s, C_2 = g^s, C_3 = h^s$.*

- *Trapdoor Queries: Taking $(w, ID_r, ID_s)$ as input, $\mathcal{B}$ randomly chooses $r \in Z_p^*$, and computes the trapdoor $T_w = (T_1, T_2)$ as follows:*

  1. *If $(ID_r, ID_s) = (ID_r^*, ID_s^*)$ or $(ID_r, ID_s) = (ID_s^*, ID_r^*)$, $\mathcal{B}$ computes $T_1 = H(\hat{e}(H_1(ID_s), H_1(ID_r)^\alpha) \cdot Z^\alpha), w) \cdot h^r$ and $T_2 = g^r$.*

  2. *Otherwise, at least one of $ID_r$ and $ID_s$ is not equal to $ID_r^*$ or $ID_s^*$. Without loss of generality, we assume that $ID_s \notin \{ID_s^*, ID_r^*\}$. $\mathcal{B}$ recovers $< (ID_s, ID_r), g^{\mu_{r,s}}, \mu_{r,s} >$ from $L_{H_2}$ and $< ID_s, PK_{ID_s} >$ from $L_{PK}$, computes $k = \hat{e}(H_1(ID_s), H_1(ID_r)^\alpha) \cdot e(g^y, PK_{ID_s})^{\mu_{r,s}}$ and returns $T_1 = H(k, w) \cdot h^r, T_2 = g^r$.*

- **Challenge**: *$\mathcal{A}_{II}$ submits two challenge keywords $w_0^*, w_1^*$, the sender's challenge identity $ID_s^*$ and the receiver's challenge identity $ID_r^*$. $\mathcal{B}$ chooses a random bit $b \in \{0, 1\}$ and an element $r \in Z_p^*$, computes ciphertext $C_1^* = \hat{e}(H(\hat{e}(H_1(ID_s^*)^\alpha, H_1(ID_r^*)) \cdot Z^\alpha), w_b^*), PK_{Csvr})^s, C_2^* = g^s, C_3^* = h^s$ and returns ciphertext $C_{w_b^*} = (C_1^*, C_2^*, C_2^*)$ to $\mathcal{A}_{II}$.*

- **Phase2**: *$\mathcal{B}$ responds as in phase 1.*

- **Guess**: *$\mathcal{A}_I$ outputs a bit $b' \in \{0, 1\}$, and then $\mathcal{B}$ outputs $\eta' = 0$ if $b' = b$ and 1 otherwise.*

*If $\mathcal{B}$ guesses that the challenge identities are incorrect, then $\mathcal{B}$ aborts. Denote by abt the event that $\mathcal{B}$ aborts. The probability that event abt does not occur is $\frac{1}{q_{H_2}}$.*

*Assume that $\mathcal{B}$ does not abort in the game. If $Z = \hat{e}(g, g)^{xyz}$, the view of $\mathcal{A}_{II}$ is the same as in a real attack, and $\mathcal{A}_{II}$ succeeds in the game with probability $Adv_{\mathcal{A}_{II}}^T + \frac{1}{2}$. If $Z$ is chosen from $\mathbb{G}_2$*

randomly, then $k$ is also a random element in $\mathbb{G}_1$; hence, $\mathcal{A}_{\mathcal{II}}$ would win Game 4 with probability at most $\frac{1}{2}$. Therefore, the advantage of $\mathcal{B}$ in solving the DBDH problem is

$$Adv_{\mathcal{B}}^{DBDH}$$

$$= |\Pr_{[\eta'=\eta|abt]} \cdot \Pr_{[abt]} + \Pr_{[\eta'=\eta|\neg abt]} \cdot \Pr_{[\neg abt]} - \frac{1}{2}|$$

$$= |\frac{1}{2}(1 - \Pr_{[\neg abt]}) + \Pr_{[\neg abt]} \cdot (\Pr_{[\eta'=0|\neg abt \wedge \eta=0]} \cdot \Pr_{[\eta=0]} + \Pr_{[\eta'=1|\neg abt \wedge \eta=1]} \cdot \Pr_{[\eta=1]}) - \frac{1}{2}|$$

$$\geq |\frac{1}{2} - \frac{1}{2} \cdot \Pr_{[\neg abt]} + \Pr_{[\neg abt]} \cdot \left(\frac{1}{2}\left(Adv_{\mathcal{A}_{\mathcal{II}}}^{T}(\lambda) + \frac{1}{2}\right) + \frac{1}{2} \cdot \frac{1}{2}\right) - \frac{1}{2}|$$

$$= \frac{1}{2}\Pr_{[\neg abt]} \cdot Adv_{\mathcal{A}_{\mathcal{II}}}^{T}(\lambda)$$

$$= \frac{1}{2} \cdot \frac{1}{q_{H_2}} \cdot dv_{\mathcal{A}}\mathcal{II}^{T}(\lambda)$$

If $Adv_{\mathcal{A}}\mathcal{II}^{T}(\lambda)$ is not negligible, then $Adv_{\mathcal{B}}^{DBDH}(\lambda)$.

**Theorem 3** *Our* dCLPAEKS *scheme satisfies designated testability under the assumption that DBDH is intractable.*

**Proof 5** *Assume that outside adversary $\mathcal{A}_{\mathcal{I}}$ tries to break the designated testability of our* dCLPAEKS *scheme. We build an algorithm $\mathcal{B}$ with $\mathcal{A}_{\mathcal{I}}$ as a subroutine to solve the DBDH problem. Given a DBDH instance $(\mathbb{G}_1, \mathbb{G}_2, \hat{e}, p, g, g^x, g^y, g^z, Z)$, $\mathcal{B}$ interacts with $\mathcal{A}_{\mathcal{I}}$ as follows:*

- **Setup**: *$\mathcal{B}$ selects $h$ from $\mathbb{G}_1$ and $\alpha \in Z_p^*$ randomly and sets params $= (\mathbb{G}_1, \mathbb{G}_2, \hat{e}, p, g, h, m_{pk} = g^\alpha)$ and $PK_{Csvr} = g^x$. Then, $\mathcal{B}$ sends params and $PK_{Csvr}$ to $\mathcal{A}_{\mathcal{I}}$.*

- **Phase 1**: *$\mathcal{B}$ answers $\mathcal{A}_{\mathcal{I}}$s inquiries as follows, assuming that $\mathcal{A}_{\mathcal{I}}$ does not repeat his inquiries.*

  - *$H_1$ Queries: Given an identity ID, $\mathcal{B}$ randomly chooses an element from $\mathbb{G}_1$ as the $H_1(ID)$ value and returns it to $\mathcal{A}_{\mathcal{I}}$.*

  - *$H_2$ Queries: Given a pair of identities $(ID_s, ID_r)$, $\mathcal{B}$ randomly selects an element from $\mathbb{G}_1$ as the $H_2(ID_s, ID_r)$ value, and outputs it.*

  - *$H$ Queries: A list $L_H$ is maintained by $\mathcal{B}$ that is initially empty. Taking an element $k$ and a keyword $w$ as input, $\mathcal{B}$ randomly chooses $\mu_{k,w} \in Z_p^*$, returns $H(k, w) = g^y g^{\mu_{k,w}}$ to $\mathcal{A}_{\mathcal{I}}$ and adds $< (k, w), H(k, w), \mu_{k,w} >$ into $L_H$.*

  - *Secret Key Queries: $\mathcal{B}$ maintains a list $L_{SK}$ that is initially empty. Taking an identity ID as input, $\mathcal{B}$ selects a random number $\beta_{ID} \in Z_p$ as the secret value of identity ID and returns the secret key $SK_{ID} = ((H_1(ID)^\alpha, \beta_{ID})$ to $\mathcal{A}_{\mathcal{I}}$. Then, $\mathcal{B}$ adds $< ID, H_1(ID)^\alpha, \beta_{ID} >$ into $L_{SK}$.*

  - *Public Key Queries: $\mathcal{B}$ maintains a list $L_{PK}$ that is initially empty. Given an identity ID, $\mathcal{B}$ recovers the tuple $< ID, H_1(ID)^\alpha, \beta_{ID} >$ from $L_{SK}$, computes the public key $PK_{ID} = (g^\alpha)^{\beta_{ID}}$, and returns it to $\mathcal{A}_{\mathcal{I}}$. Then, $< ID, PK_{ID} >$ is added to $L_{PK}$.*

- **Challenge**: *$\mathcal{A}_{\mathcal{I}}$ submits two different challenge keywords $w_0^*, w_1^*$, the sender's identity $ID_s^*$ and the receiver's identity $ID_r^*$. $\mathcal{B}$ chooses a random bit $b \in \{0, 1\}$ and recovers the tuple*

  *$< (k^*, w_b^*), H((k^*, w_b^*), \mu_{k^*, w_b^*} >$, where $k^* = \hat{e}(SK_{ID_s^*}^1, H_1(ID_r^*)) \cdot \hat{e}(H_2(ID_s^*, ID_r^*), PK_{ID_r^*})^{SK_{ID_s^*}^2}$. If there is no such tuple, $\mathcal{B}$ generates it as in previous queries. Then, $\mathcal{B}$ computes the challenge ciphertext $C_{w_b^*} = (C_1^*, C_2^*)$, where $C_1^* = Z \cdot \hat{e}(g^z, g^x)^{\mu_{k^*, w_b^*}}, C_2^* = g^z, C_3^* = h^z$.*

- **Phase 2**: *The simulator $\mathcal{B}$ responds as in phase 1.*

**Table 1. Comparison of security.**

| Scheme | Functionalities | | | |
|---|---|---|---|---|
| | C Ind | T Ind | SCF | IKGAs |
| SCF-MCLPEKS⁺ [17] | √ | × | √ | × |
| CLPEKS [43] | √ | × | √ | × |
| SCF-MCLPEKS [44] | √ | × | √ | × |
| CL-dPAEKS [19] | √ | × | √ | × |
| Proposed scheme | √ | √ | √ | √ |

C Ind: Ciphertext Indistinguishability, T Ind: Trapdoor Indistinguishability, SCF: Secure Channel Free, IKGAs: Security against Inside Keyword Guessing Attacks.

- **Guess**: $\mathcal{A}_{\mathcal{I}}$ outputs a bit $b'$; if $b' = b$, B outputs $\eta' = 0$, otherwise 1. If $Z = \hat{e}(g,g)^{xyz}$, the view of $\mathcal{A}_{\mathcal{I}}$ is the same as in a real attack, and $\mathcal{A}_{\mathcal{I}}$ succeeds in Game 5 with probability $Adv^D_{\mathcal{A}_{\mathcal{I}}} + \frac{1}{2}$. If $Z$ is selected from $\mathbb{G}_2$ randomly, then $C_1^*$ is also a random element in $\mathbb{G}_1$; hence, $\mathcal{A}_{\mathcal{I}}$ succeeds in the game with probability at most $\frac{1}{2}$. Therefore, the advantage of B in solving the DBDH problem is

$$Adv^{DBDH}_{\mathcal{B}}$$
$$= |\Pr_{[\eta'=\eta|\eta=1]} \cdot \Pr_{[\eta=1]} + \Pr_{[\eta'=\eta|\eta=0]} \cdot \Pr_{[\eta=0]} - \frac{1}{2}|$$
$$= |\frac{1}{2} \cdot \frac{1}{2} + \frac{1}{2}\left(Adv^D_{\mathcal{A}_{\mathcal{I}}}(\lambda) + \frac{1}{2}\right) - \frac{1}{2}|$$
$$= \frac{1}{2} \cdot Adv^D_{\mathcal{A}_{\mathcal{I}}}$$

If $Adv^D_{\mathcal{A}_{\mathcal{I}}}(\lambda)$ is not negligible, then $Adv^{DBDH}_{\mathcal{B}}$.

## Evaluation

In this section, we evaluate the security properties, computational complexity and communication overhead of our scheme, and compare it with the schemes proposed in [17, 43, 44], and [19]. Table 1 shows the comparison between these schemes and our dCLPAEKS scheme in terms of security. As shown by Table 1, our scheme provides security against inside keyword

**Table 2. Comparison of communication overhead.**

| Scheme | communication overhead | | |
|---|---|---|---|
| | Size(PK) | Size(C) | Size($T_w$) |
| SCF-MCLPEKS⁺ [17] | $|G_1|$ | $2|G_1| + |G_2|$ | $2|G_1|$ |
| CLPEKS [43] | $|G_1|$ | $|G_1| + |Z_p|$ | $|G_1|$ |
| SCF-MCLPEKS [44] | $2|G_1|$ | $|G_1| + |Z_p|$ | $|G_1|$ |
| CL-dPAEKS [19] | $2|G_1|$ | $2|G_1|$ | $2|G_1| + |G_2|$ |
| Proposed scheme | $|G_1|$ | $2|G_1| + |G_2|$ | $2|G_1|$ |

Size(PK): Size of Public Key, Size(C): Size of Ciphertext, Size($T_w$): Size of Trapdoor, $|G_1|$: Size of an element in $G_1$, $|G_2|$: Size of an element in $G_2$, $|Z_p|$: Size of an element in $Z_p$.

**Table 3. Computational efficiency comparison.**

| scheme | PEKS | Trapdoor | Test |
|---|---|---|---|
| SCF-MCLPEKS⁺ [17] | 4E+2H+h+2P | 3E+h+2A | E+2P+h+A+M |
| CLPEKS [43] | 5E+3P+3H+2h | 3E+H+h+2A | 2E+P+h+4A |
| SCF-MCLPEKS [44]) | 4E+3P+3H+h+A | E+H+A | E+P+2H+h+2A |
| CL-dPAEKS [19] | 5E+H+2h+3A | 7E+H+3h+P+4A | 4E+2P+M |
| Our scheme | 3E+3H+3P | 3E+3H+2P+M | E+2P+M |

E: a scalar multiplication operation; P: a bilinear pairing operation; H: a Hash-to-point operation; h: a general hash function operation; A: an addition operation; M: a multiplication operation

guessing attacks and against outside keyword guessing attacks without requiring a secure channel.

The communication overhead of the five schemes is given in Table 2. According to Table 2, the communication overhead of our dCLPAEKS scheme is almost the same as that of SCF-MCLPEKS⁺ and CL-dPAEKS.

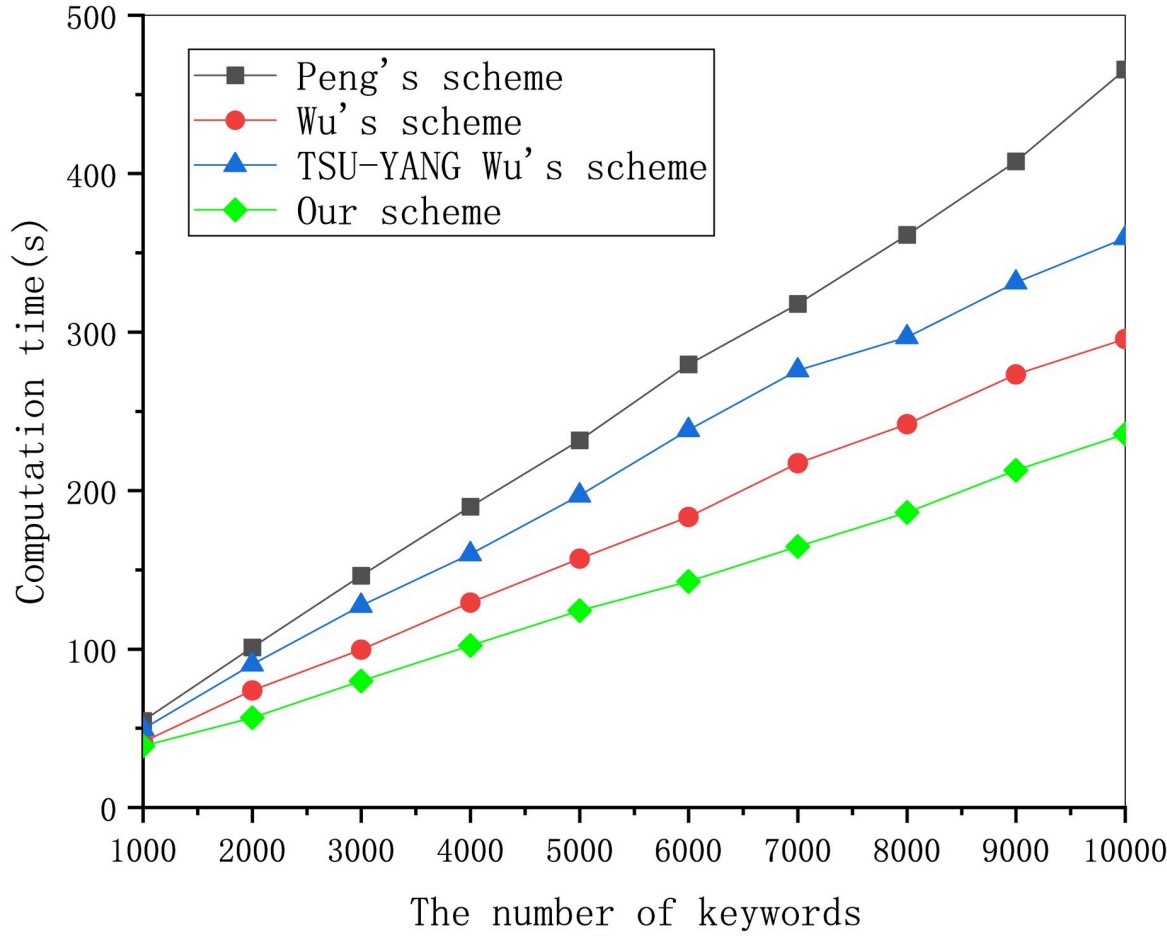

**Fig 1.**

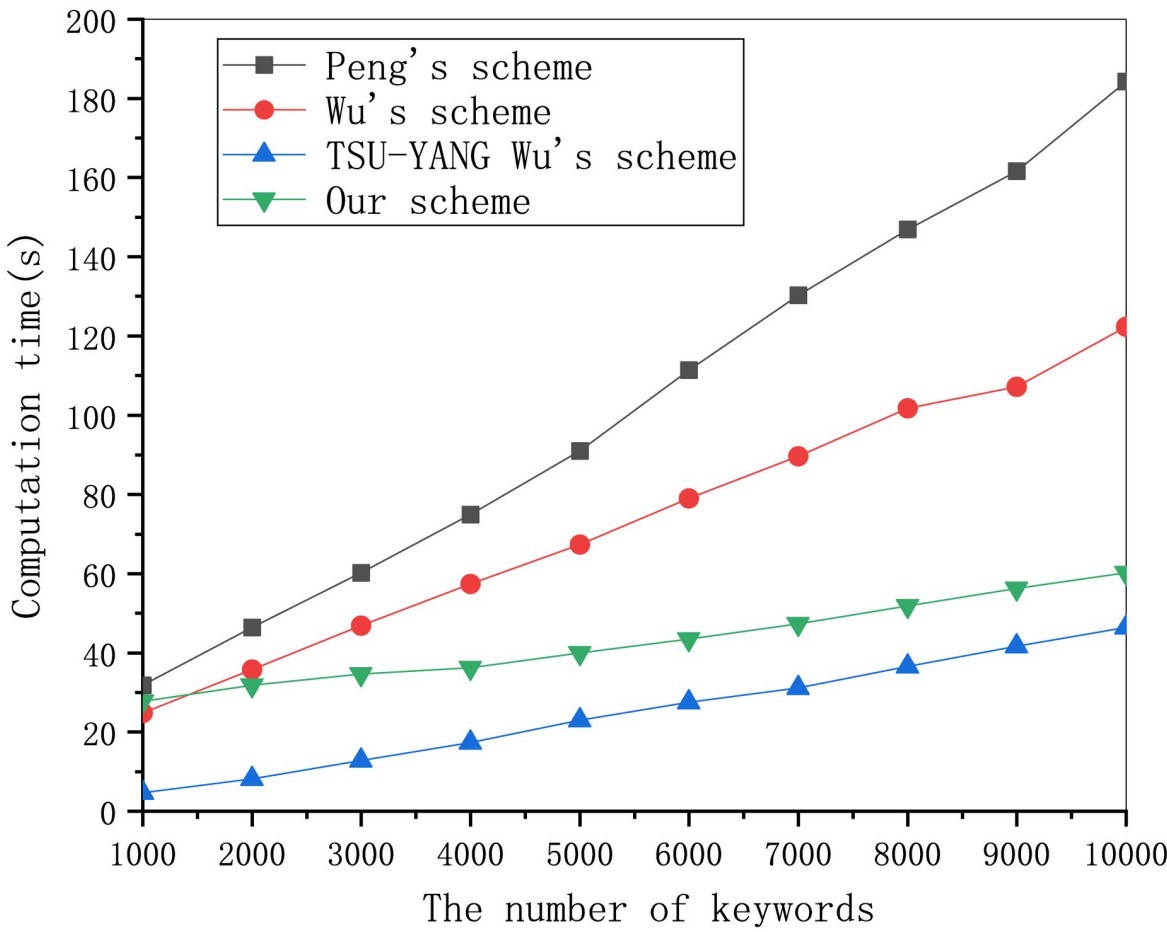

**Fig 2.**

The comparison of computational complexity is given in Table 3. We compare the computational costs of the PEKS, Trapdoor and Test algorithms of the schemes. The results show that our scheme is comparable with the other schemes.

We implemented our scheme, the SCF-MCLPEKS$^+$ scheme [17], the CL-dPAEKS scheme [19] and CLPEKS [43] on a laptop with a 3.10-GHz Intel i5 CPU with a 64-GB memory and an Ubuntu Linux operating system. We used the PBC library [48], in which Type-A pairing was chosen. The pairing operation is based on the curve $y^2 = x^3 + x$ over the field $F_p$. The parameter set is $|G_1| = |G_2| = 128$-bit.

To compare the computational efficiency of the four schemes, including our scheme in more detail, we tested the running time of the ciphertext algorithm and trapdoor algorithm. As shown in Fig 1, the ciphertext generation of our scheme has the highest computational efficiency compared with the other schemes. According to Fig 2, as the number of keywords increases, our scheme outperforms the other two schemes in computational efficiency. Fig 1. Computation cost of ciphertext generation in different schemes. Fig 2. Computation cost of trapdoor generation in different schemes.

## Conclusions

In a cloud-based IoT environment, protecting the privacy and security of sensitive data stored in the cloud is a major concern. An effective method is certificateless public key searchable

encryption (CLPEKS), which both enables search over encrypted data and avoids the problems of certificate management and key escrow. In this paper, we demonstrated that the security reduction for the CLPAEKS scheme proposed by He et al. is incorrect under two types of adversaries, and Ma et al.'s CLPEKS scheme is susceptible to an off-line KGAs. We then proposed a new certificateless public key searchable encryption scheme, which overcomes a limitation of these two schemes—the need for a secure channel—and solves the security defect that the CLPEKS scheme cannot resist a KGAs. In addition, in comparison with the other recently proposed CLPEKS schemes, the performance analysis demonstrates that our scheme is more efficient and has higher security.

## Author Contributions

**Conceptualization:** Bin Wu, Caifen Wang, Hailong Yao.

**Formal analysis:** Bin Wu, Caifen Wang.

**Methodology:** Bin Wu, Caifen Wang, Hailong Yao.

**Resources:** Bin Wu, Caifen Wang.

**Writing – original draft:** Bin Wu, Caifen Wang.

**Writing – review & editing:** Bin Wu, Caifen Wang.

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
