## [Decision Letter · Decision Letter 0]

2 Jan 2020

PONE-D-19-34208

Security Analysis and Secure Channel Free Certificateless Searchable Public Key Authenticated Encryption for cloud-based Internet of Things

PLOS ONE

Dear Dr Wu,

Thank you for submitting your manuscript to PLOS ONE. After careful consideration, we feel that it has merit but does not fully meet PLOS ONE’s publication criteria as it currently stands. Therefore, we invite you to submit a revised version of the manuscript that addresses the points raised during the review process.

We would appreciate receiving your revised manuscript by Feb 16 2020 11:59PM. To enhance the reproducibility of your results, we recommend that if applicable you deposit your laboratory protocols in protocols.io, where a protocol can be assigned its own identifier (DOI) such that it can be cited independently in the future. For instructions see: http://journals.plos.org/plosone/s/submission-guidelines#loc-laboratory-protocols

We look forward to receiving your revised manuscript.

Kind regards,

He Debiao

Academic Editor

PLOS ONE

Journal Requirements:

2.

We suggest you thoroughly copyedit your manuscript for language usage, spelling, and grammar. If you do not know anyone who can help you do this, you may wish to consider employing a professional scientific editing service.  

3. We note that Figure(s) [1-3] in your submission contain copyrighted images. All PLOS content is published under the Creative Commons Attribution License (CC BY 4.0), which means that the manuscript, images, and Supporting Information files will be freely available online, and any third party is permitted to access, download, copy, distribute, and use these materials in any way, even commercially, with proper attribution. For more information, see our copyright guidelines: http://journals.plos.org/plosone/s/licenses-and-copyright.

1.    You may seek permission from the original copyright holder of Figure(s) [1-3] to publish the content specifically under the CC BY 4.0 license.

Reviewers' comments:

Reviewer's Responses to Questions

**Comments to the Author**

1. Is the manuscript technically sound, and do the data support the conclusions?

Reviewer #1: Yes

Reviewer #2: Yes

2. Has the statistical analysis been performed appropriately and rigorously? 

Reviewer #1: Yes

Reviewer #2: Yes

3. Have the authors made all data underlying the findings in their manuscript fully available?

Reviewer #1: Yes

Reviewer #2: Yes

4. Is the manuscript presented in an intelligible fashion and written in standard English?

Reviewer #1: Yes

Reviewer #2: Yes

5. Review Comments to the Author

Reviewer #1: In this paper, the authors have demonstrated that the security reduction for the CLPAEKS scheme proposed by He et al. is incorrect under two types of adversaries, and they have also pointed that Ma et al.’s CLPEKS scheme is susceptible to an offline KGA. Furthermore, they have proposed a new certificateless public key searchable encryption scheme, which overcomes a limitation of these two schemes—the need for a secure channel—and solves the security defect that the CLPEKS scheme cannot resist a KGA. Moreover, in comparison with the other CLPEKS schemes, the performance analysis demonstrates that their scheme has higher security and comparable efficiency

· The English writing should be carefully checked before it can be accepted in the Journal.

· More related work about cloud-based Internet of Things should be cited in the manuscript.

Additional Questions:

Does the paper contribute to the body of knowledge?: Yes. The authors proposed a new certificateless public key searchable encryption scheme which has higher retrieval efficiency and wider retrieval range.

Is the paper technically sound?: Yes. The paper presents the modeling of the solution clearly and the security proof of the specific scheme is given. Besides, the paper presents conducted experimental evaluation with preliminary interesting results.

Is the subject matter presented in a comprehensive manner?: Yes, it is. The paper is well organized: it presents the problem, the proposed and modeling of solution, and finally, an experimental evaluation.

Are the references provided applicable and sufficient?: No. the references should be enhanced.

Reviewer #2: This paper studies certificateless public key searchable encryption schemes (CLPAEKS), which is an interesting and relevant primitive to achieve confidentiality and security in the area of outsourced computing. They give security analysis on two previous proposed CLPAES, and show the security flaws of two previous schemes by He et al (IEEE T IND INFORM. 2018) and Ma et al (COMPUT ELECTR ENG. 2018). They also propose a channel-free certificateless searchable public key authenticated encryption (dCLPAEKS) scheme and prove that it is secure against inside keyword guessing attacks under the enhanced security model.

The paper is clearly written and the proofs and analysis seem correct.

6. PLOS authors have the option to publish the peer review history of their article (what does this mean?). If published, this will include your full peer review and any attached files.

Reviewer #1: No

Reviewer #2: No

---

## [Author Response · Author response to Decision Letter 0]

28 Feb 2020

Original Manuscript ID: PONE-D-19-34208 

Original Article Title: “Security Analysis and Secure Channel Free Certificateless Searchable Public Key Authenticated Encryption for cloud-based Internet of Things ”

To: PLOS ONE Editor

Re: Response to reviewers

Dear Editor,

Thank you for allowing a resubmission of our manuscript, with an opportunity to address the reviewers’ comments.

We are uploading (a) our point-by-point response to the comments (below) (response to reviewers), (b) an updated manuscript with yellow highlighting indicating changes, and (c) a clean updated manuscript without highlights (PDF main document), (d) a certificate issued by American Journal Specialist (AJE) to help us with language editing.

Best regards,

Bin Wu, Caifen Wang, Hailong Yao.

Journal Requirements, Concern # 1:  Please ensure that your manuscript meets PLOS ONE's style requirements, including those for file naming. The PLOS ONE style templates can be found at http://www.journals.plos.org/plosone/s/file?id=wjVg/PLOSOne_formatting_sample_main_body.pdf and http://www.journals.plos.org/plosone/s/file?id=ba62/PLOSOne_formatting_sample_title_authors_affiliations.pdf.

Author response: We have carefully revised our manuscript completely according to the PLOS ONE style templates.

Author action: We carefully modified the non-compliant parts of the manuscript according to the style template of PLOS ONE.

Journal Requirements, Concern # 2: We suggest you thoroughly copyedit your manuscript for language usage, spelling, and grammar. If you do not know anyone who can help you do this, you may wish to consider employing a professional scientific editing service. 

Author response: We have carefully revised our manuscript completely according to the suggestion.

Author action: We have selected American Journal Experts (AJE) to help us with language editing to ensure that our manuscripts meet journal submission guidelines. This certificate was issued on January 23, 2020 and may be verified on the AJE website using the verification code FFB0-56B0-5224-C035-2CB8 .

Journal Requirements, Concern # 3: We note that Figure(s) [1-3] in your submission contain copyrighted images. We require you to either (1) present written permission from the copyright holder to publish these figures specifically under the CC BY 4.0 license, or (2) remove the figures from your submission

Author response: We have carefully revised our manuscript completely according to the suggestion.

Author action: Since the copyright-protected image material contained in our submitted Figures [1-3] was downloaded from the Internet, it was not easy to obtain permission from the original copyright holder to publish these figures under the CC BY 4.0 license. So we use three paragraphs to describe the meaning of Figure [1-3] in detail respectively , thus remove the figure [1-3].

Reviewer#1, Concern # 1: The English writing should be carefully checked before it can be accepted in the Journal.

Author response: We have carefully revised our manuscript completely according to the suggestion.

Author action: We updated the manuscript by modifying the language usage, spelling, and grammar, following the language editing suggestions provided by American Journal Experts (AJE). This certificate was issued on January 23, 2020 and may be verified on the AJE website using the verification code FFB0-56B0-5224-C035-2CB8 .

Reviewer#2, Concern # 2: More related work about cloud-based Internet of Things should be cited in the manuscript. 

Author response: We have carefully revised our manuscript completely according to the suggestion.

Author action: We carefully studied the related work in 2017, 2018 and 2019, selected 6 representative works[9]-[14] from them, introduced them in the introduction, and added them into the reference. 

REFERENCES:

[9] Babu S M, Lakshmi A J, Rao B T et al. A study on cloud based Internet of Things: CloudIoT. In: Proc. GCCT 2015. 60-65. doi:10.1109/GCCT.2015.7342624.

[10] Conti M, Dehghantanhab A, Frankec K, Watsond S. Internet of Things security and forensics: Challenges and opportunities. Future Generation Computer Systems. 2018, 78:544-546. doi: 10.1016/j.future.2017.07.060.

[11] Ojha T, Misra S, Raghuwanshi R N, Poddar H. DVSP: Dynamic Virtual Sensor Provisioning in Sensor Cloud-Based Internet of Things. IEEE Internet of Things Journal. 2019, 6(3):5265-5272. doi: 10.1109/JIOT.2019.2899949.

[12] Pan W, Chai C. Structure-aware Mashup service Clustering for cloud-based Internet of Things using genetic algorithm based clustering algorithm. Future Generation Computer Systems. 2018, 87:267-277. doi:10.1016/j.future.2018.04.052.

[13] Meerja K A, Naidu P V, Kalva S R K. Price Versus Performance of Big Data Analysis for Cloud Based Internet of Things Networks. Mobile Netw Appl. 2019, 24:1078-1094. doi: 10.1007/s11036-018-1063-6.

[14] Boveiri H R, Khayami R, Elhoseny M et al. An efficient Swarm-Intelligence approach for task scheduling in cloud-based internet of things applications. J Ambient Intell Human Comput. 2019, 10:3469-3479. doi: 10.1007/s12652-018-1071-1.

---

## [Decision Letter · Decision Letter 1]

9 Mar 2020

Security Analysis and Secure Channel Free Certificateless Searchable Public Key Authenticated Encryption for cloud-based Internet of Things

PONE-D-19-34208R1

Dear Dr. Wu,

We are pleased to inform you that your manuscript has been judged scientifically suitable for publication and will be formally accepted for publication once it complies with all outstanding technical requirements.

With kind regards,

He Debiao

Academic Editor

PLOS ONE

Additional Editor Comments (optional):

Reviewers' comments:

Reviewer's Responses to Questions

**Comments to the Author**

1. If the authors have adequately addressed your comments raised in a previous round of review and you feel that this manuscript is now acceptable for publication, you may indicate that here to bypass the “Comments to the Author” section, enter your conflict of interest statement in the “Confidential to Editor” section, and submit your "Accept" recommendation.

Reviewer #1: All comments have been addressed

Reviewer #2: All comments have been addressed

2. Is the manuscript technically sound, and do the data support the conclusions?

Reviewer #1: Yes

Reviewer #2: Yes

3. Has the statistical analysis been performed appropriately and rigorously? 

Reviewer #1: Yes

Reviewer #2: Yes

4. Have the authors made all data underlying the findings in their manuscript fully available?

Reviewer #1: Yes

Reviewer #2: Yes

5. Is the manuscript presented in an intelligible fashion and written in standard English?

Reviewer #1: Yes

Reviewer #2: Yes

6. Review Comments to the Author

Reviewer #1: The authors have responded all the comments and I suggest to accept the paper to be of publication in this journal.

Reviewer #2: The authors have revised the paper carefully and my previous comments have been addressed properly and I recommend for acceptance.

7. PLOS authors have the option to publish the peer review history of their article (what does this mean?). If published, this will include your full peer review and any attached files.

Reviewer #1: No

Reviewer #2: No

---

## [Editor Report · Acceptance letter]

20 Mar 2020

PONE-D-19-34208R1 

Security analysis and secure channel-free certificateless searchable public key authenticated encryption for a cloud-based Internet of Things 

Dear Dr. Wu:

I am pleased to inform you that your manuscript has been deemed suitable for publication in PLOS ONE. Congratulations! Your manuscript is now with our production department. 

With kind regards,

on behalf of

Dr. He Debiao 

Academic Editor

PLOS ONE